# Retrieval-based Disentangled Representation Learning with Natural Language Supervision

**Jiawei Zhou**[1]  **Xiaoguang Li**[2]  **Lifeng Shang**[2]  **Xin Jiang**[2]  **Qun Liu**[2]  **Lei Chen**[1]

The Hong Kong University of Science and Technology[1]
Huawei Noah's Ark Lab[2]
{jzhoubu,leichen}@ust.hk
{lixiaoguang11,Shang.Lifeng,Jiang.Xin,qun.liu}@huawei.com

## Abstract

Disentangled representation learning remains challenging as the underlying factors of variation in the data do not naturally exist. The inherent complexity of real-world data makes it unfeasible to exhaustively enumerate and encapsulate all its variations within a finite set of factors. In light of this, we present Vocabulary Disentangled Retrieval (VDR), a retrieval-based framework that harnesses natural language as proxies of the underlying data variation to drive disentangled representation learning. Our approach employs a bi-encoder model to represent both data and natural language in a vocabulary space, enabling the model to distinguish dimensions that capture intrinsic characteristics within data through its natural language counterpart, thus facilitating disentanglement. We extensively assess the performance of VDR across 15 retrieval benchmark datasets, covering text-to-text and cross-modal retrieval scenarios, as well as human evaluation. Our experimental results compellingly demonstrate the superiority of VDR over previous bi-encoder retrievers with comparable model size and training costs, achieving an impressive 8.7% improvement in NDCG@10 on the BEIR benchmark, a 5.3% increase on MS COCO, and a 6.0% increase on Flickr30k in terms of mean recall in the zero-shot setting. Moreover, the results from human evaluation indicate that interpretability of our method is on par with SOTA captioning models. [1]

## 1 Introduction

Disentangled representation learning (Bengio et al., 2009; 2013; Higgins et al., 2017; Chen et al., 2018) aims to identify the underlying factors of variations within data and correlate them to distinct units of the learned representation. Essentially, a well-disentangled representation independently captures underlying factors that explain the data, thereby facilitating explainability, controllability, and debugability of machine learning. However, as pointed out by Bengio et al. (2013), a fundamental challenge lies in defining a set of factors that are sufficiently informative to represent the data while remaining independent of the tasks at hand. Despite extensive research efforts spanning several years, the journey to effectively address this challenge remains an ongoing endeavor.

Given the absence of an universal set of factors of variations behind data, supervising disentanglement remains challenging. Early works (Higgins et al., 2017; Chen et al., 2016; Kim & Mnih, 2018) employ autoencoder framework without supervision, aiming to impose constraints on VAE (Kingma & Welling, 2013) to enhance the independence among latent factors, without explicitly defining their meaning. These methods are later challenged by Locatello et al. (2019a), revealing that unsupervised disentanglement is highly reliant on inductive biases, randomness, and parameter choices. Subsequent research directions have shifted toward incorporating varying degrees of supervision. For instance, certain research (Locatello et al., 2019b; Gabbay et al., 2021) explored supervising disentanglement using a small subset of labeled data from synthetic toy datasets. These efforts often revolve around tasks of controllable image generation, typically focusing on synthetic images (Reed et al., 2015; Matthey et al., 2017) with limited explanatory factors such as object shape, color, and

---

[1]The code is available at: https://github.com/jzhoubu/VDR

category. Nevertheless, the feasibility of applying these task-specific explanatory factors to real-world scenarios remains constrained. Another line of research employ relational information among data samples, such as group-wise (Locatello et al., 2020; Shakerinava et al., 2022), pair-wise (Chen & Batmanghelich, 2020), sequence (Bai et al., 2021; Li et al., 2022b; Miyato et al., 2022) and graph-based (Ma et al., 2019; Mercatali et al., 2022; Wang et al., 2022c) information, to weakly supervise disentanglement learning. However, acquiring annotations or structured data for these methods can pose challenges, potentially limiting their generalizability within real-world scenarios.

It is important to recognize that real-world data is inherently complex and cannot be fully encapsulated by a limited set of attributes. In practice, individuals often resort to natural language to describe and identify real-world objects, as it offers a versatile way to represent their diverse characteristics. These natural language counterparts can be tokenized into a finite vocabulary space, effectively serving as a proxy for capturing the inherent variation in the data.

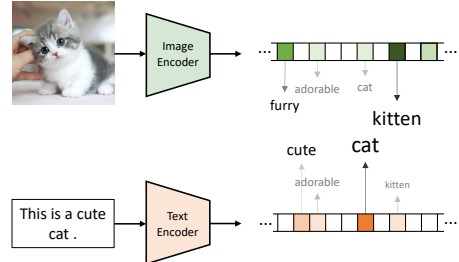

Figure 1: Illustration of retrieval-based framework. The color intensity reflects the higher values along the dimension.

Motivated by this, we present Vocabulary Disentangled Retrieval (VDR), a simple yet effective retrieval-based approach to drive disentangled representation learning. In essence, VDR represents both data and their linguistic counterparts within a $|\mathbb{V}|$-dimensional lexical representation space, where each dimension corresponds to a token from the vocabulary $\mathbb{V}$. The value along each dimension quantifies the semantic relevance between the input data and the corresponding tokens. By aligning the sparse representations of the data to those of their linguistic counterparts, VDR encourages the dimension-wise disentanglement on the lexical representations. Our methodology is thoroughly evaluated across 15 datasets, covering both text-to-text and cross-modal retrieval, as well as human evaluations. Experimental results demonstrate the consistent superiority of VDR over existing retrieval baselines with similar complexity, in terms of both model size and training configurations. Furthermore, VDR achieves competitive results when compared to more sophisticated retrieval methods that employ intricate techniques to reach state-of-the-art performance.

In summary, our work technically improves the current lexical retriever and extends its functionality to handle multi-modal data. We utilize this progress to drive disentangled representation learning in the context of multi-modal data, with a primary focus on contributing to the broader field of explainable machine learning research. Our contributions can be categorized into two aspects. From disentangled representation learning aspects:

1. We showcase the feasibility of harnessing natural language as a source of supervision for disentangled representation learning. Our methodology considers a task-agnostic vocabulary as explanatory variables, highlighting its wide-ranging applicability and adaptability.

2. Through both human evaluations and retrieval benchmarks, we establish that the representations learned by our methodology exhibit rational dimensional values to capture the intrinsic characteristics of input data.

From sparse lexical retrieval aspects:

1. We pioneer the viability of embedding non-textual data into lexical representations, expanding the scope of lexical retrievers to encompass multi-modal data from diverse datasets and tasks. This extension presents distinctive challenges, which we have identified and addressed through innovative solutions.

2. Our approach demonstrates notable improvements in both text-to-text retrieval and cross-modal retrieval. It significantly surpasses existing dense and sparse retrievers with comparable training cost while maintaining competitiveness with more sophisticated and advanced retrieval baselines.

3. Notably, our retriever supports nonparametric inference, eliminating the need of neural network forwarding during inference. This enhancement results in over a 10x improvement in efficiency while maintaining strong effectiveness, making it particularly valuable for low-resource applications.

## 2 BACKGROUND

### 2.1 INFORMATION RETRIEVAL

Information retrieval (Manning, 2009; Mitra et al., 2018; Zhao et al., 2022) aims to find specific targets $p$ that fulfill certain information needs from a vast corpus based on a given query $q$. This is typically achieved by utilizing the bi-encoder framework, which employs two independent encoders to encode the query and target into vectors and measures their relevance as the inner product of their representations:

$$sim(q, p) = E_q(q) \cdot E_p(p)^{\mathrm{T}}$$

where $sim(q, p) \in \mathbb{R}$ is the similarity between $q$ and $p$, and $E_q(\cdot)$, $E_p(\cdot)$ are query encoder and target encoder, respectively.

**Notation**   We use $q$ to indicate the textual query, while $p$ indicates the target, which could be either text or an image. The variable $x$ encompasses both query and target data. Whenever we refer to input data $x$, we imply that both the query and target sides undergo the same processing pipeline.

**Dense Retrieval**   Dense retrieval (Karpukhin et al., 2020) stands as a prominent technique in retrieval applications, wherein the encoder embed the input $x$ into a $d$-dimensional latent representation, i.e., $E_{dense}(x) \in \mathbb{R}^d$.

**Lexical Retrieval**   Lexical retrievers (Bai et al., 2020; Formal et al., 2021b) encode the data into a sparse lexical representation denoted as $E_{lexical}(x) = V(x) \odot G(x)$ where $\odot$ is element-wise multiplication. Here, $V : x \to \mathbb{R}^{|\mathbb{V}|}$ is a weighting function that encodes input data $x$ into a $|\mathbb{V}|$-dimensional vector with positive values. These values signify the semantic relevance between tokens in the vocabulary and the input $x$. On the other hand, $G : x \to \{0, 1\}^{|\mathbb{V}|}$ is a gating function that generates binary vector for sparsification.

### 2.2 DIMENSION-WISE SUPERVISION ON SPARSE REPRESENTATION

The primary goal of lexical retriever is to learn a weighted distribution on vocabulary to represent the input data. While there is no ground truth for such distributions in nature, the dimension-wise supervision is indirectly derived from a combination of contrastive learning and the gating function.

Consider a batch $B = \langle q_i, p_i^+, p_i^- \rangle_{i=1}^N$ comprising $N$ instances. Each instance consists of a query $q$, a positive target $p^+$, and a negative target $p^-$. The retriever training objective is based on contrastive learning (Jaiswal et al., 2020), which aims to maximize the similarity of positive pairs, denoted as $sim(q_i, p_i^+)$ for all instances $i$ in the batch, while minimize the similarity of all negative pairs, denoted as $sim(q_i, p)$ for all other target $p$ in the batch except the positive one $p_i^+$. The similarity between sparse representations can be elaborated as follows:

$$sim(q, p) = V_q(q) \odot G_q(q) \cdot (V_p(p) \odot G_p(p))^{\mathrm{T}} = \sum_{i=1}^{|\mathbb{V}|} V_q(q)[i] \cdot V_p(p)[i] \cdot \alpha_i(q, p) \tag{1}$$
$$\alpha_i(q, p) = \mathbb{I}[G_q(q)[i] = 1, G_p(p)[i] = 1]$$

where $V(x)[i]$ and $G(x)[i]$ refers to the $i$-th dimensional value within $V(x)$ and $G(x)$, respectively. The notation $\alpha_i(q, p)$ indicates whether both $G_q(q)$ and $G_p(p)$ have activated the $i$-th dimensions. In the rest of thr paper, we refer to $\alpha_i(q, p) = 1$ as the "co-activation" on dimension $i$.

In essence, the co-activation $\alpha_i(q, p)$ determines which dimensions should contribute to increasing $sim(q, p)$ whereas the contrastive objective governs whether the similarity $sim(q, p)$ should be maximized or minimized. When $\alpha_i(q, p) = 1$, the $i$-th dimension adds a positive term $V_q(q)[i] \cdot V_p(p)[i]$ to $sim(q, p)$. Depending on the objective, the retriever may either increase or decrease the values of $V_q(q)[i]$ and $V_p(p)[i]$ accordingly for optimization. When $\alpha_i(q, p) = 0$, the mechanism fails to provide direct supervision on the $i$-th dimension as the sparsification prevents the dimensional values from contributing to the contrastive objective. To summarize, the key to induce valid dimension-wise supervision hinges on the gating function $G$. It should activate dimensions that effectively represent the data, thereby enabling the co-activation $\alpha_i(q, p)$ to accurately signify the relevance between $q$ and $p$.

## 3 VOCABULARY DISENTANGLED RETRIEVER

### 3.1 MODEL ARCHITECTURE

In summary, VDR is a sparse lexical bi-encoder framework that encode the input data $x$ into sparse lexical representations:

$$E(x) = V(x) \odot G(x) \tag{2}$$

The weighting function $V = E_{base} \circ f_{dst}$ involves a conventional transformer-based encoder to encode input $x$ into a sequence of hidden states, denote as $E_{base} : x \rightarrow \mathbb{R}^{d \times L}$, and a disentanglement (DST) head to transform these latent states into a lexical representation, denoted as $f_{dst} : \mathbb{R}^{d \times L} \rightarrow \mathbb{R}^{|\mathbb{V}|}$. $L$ indicates the number of patches for images or number of tokens for text.

**Vocabulary** We adopt the vocabulary from the BERT (Devlin et al., 2019) tokenizer and discard the unused tokens, resulting in a vocabulary $\mathbb{V}$ with a size of $|\mathbb{V}|$=29522. Upon analysis employing the PyEnchant [2] library, we find that more than 20,000 of these tokens are accurately spelled. This observation indicates that most of the tokens can serve as interpretable 1-gram concepts.

**Base Encoder** The base encoders $E_{base} : x \rightarrow \mathbb{R}^{d \times L}$ transform input data into sequences of $d$-dimensional hidden states. For text input, we employ a pre-trained BERT based model as the base encoder. For image input, we use ViT-B/32 architecture (Dosovitskiy et al., 2020) and train from scratch.

**Disentanglement (DST) Head** The disentanglement head process, $f_{dst} : \mathbb{R}^{d \times L} \rightarrow \mathbb{R}^{|\mathbb{V}|}$, involves several steps. To begin, a layer normalization is applied to the output hidden states. Then, these hidden states are projected into $|\mathbb{V}|$-dimensional representations using a linear projection layer denoted as $W : \mathbb{R}^d \rightarrow \mathbb{R}^{|\mathbb{V}|}$. Subsequently, an $elu1p$ activation is employed to transform the representation into positive values. The $elu1p$ activation is defined as:

$$elu1p(x) = \begin{cases} x + 1 & \text{if } x >= 0 \\ e^x & \text{otherwise} \end{cases} \tag{3}$$

Last, following Formal et al. (2021a), a max pooling is employed to aggregate the fine-grained (token or patch) representations into a global representation, transforming the representation from $\mathbb{R}^{|\mathbb{V}| \times L}$ to $\mathbb{R}^{|\mathbb{V}|}$.

**Gating Function** Our gating function $G$ activates the dimensions corresponding to the tokens that exist in $x$ along with the top-$k$ dimensions of $V(x)$. These top-$k$ dimensions encompass tokens that not present in $x$, which can be viewed as an expansion of $x$ based on the learned weighting distribution.

$$G(x)[i] = \begin{cases} 1 & \text{if } \text{top}_k(V(x))[i] = 1 \text{ or } \text{bow}(x)[i] = 1 \\ 0 & \text{else} \end{cases} \tag{4}$$

Here, $\text{top}_k(V(x))$ is a binary vector that indicates dimensions possessing the $k$ largest values within $V(x)$, and $\text{bow}(x)$ is a bag-of-words representation of $x$. Specifically for non-textual data $x$, $\text{bow}(x)[i] = 0$ for all $i$.

### 3.2 MODEL TRAINING

The main component of our loss function is a symmetric cross-entropy (SCE) loss, defined as follows:

$$L = - \log \underbrace{\frac{\exp(sim(q_i, p_i^+)/\tau)}{\sum_{j=1}^N \exp(sim(q_i, p_j^+)/\tau) + \exp(sim(q_i, p_j^-)/\tau)}}_{\text{q-to-p}} - \log \underbrace{\frac{\exp(sim(p_i^+, q_i)/\tau)}{\sum_{j=1}^N \exp(sim(p_i^+, q_j)/\tau)}}_{\text{p-to-q}} \} \tag{5}$$

where $\tau$ is a temperature parameter.

The final loss comprises two term. The first term applies SCE loss to $V_q(q) \odot G_q(q)$ and $V_p(p)$. The second term involves the SCE loss between the nonparametric query representation $\text{bow}(q)$ and $V_p(p)$. The final loss is computed as the sum of these two terms. Further information can be found in Figure 2. The upcoming section will delve into the rationale behind this loss design.

---

[2]https://pyenchant.github.io/pyenchant/

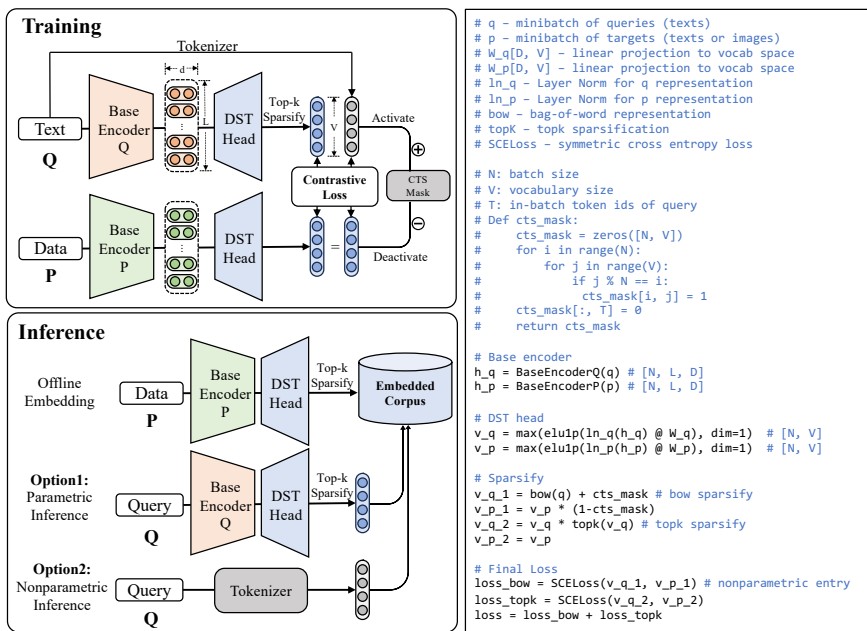

Figure 2: Left: training and inference pipeline of VDR. Right: pseudo code for training VDR.

## 3.3 EXTENDING TO CROSS-MODAL SCENARIOS

Lexical retrievers largely rely on pre-trained masked language models (MLM) (Devlin et al., 2019; Sanh et al., 2019) which undergo pre-training tasks to output probability distributions over vocabulary to predict masked tokens within a given context. This process establishes a robust basis for the initial weighting and gating distributions of a lexical retriever. As a result, the co-activation $\alpha_i(q, p)$ can effectively capture the relevance between $q$ and $p$, and the learning process will gradually shape more rational distributions. However, challenges arise in cross-modal scenarios, where the image encoder and its projection layer need to be trained from scratch. In such cases, random weighting and gating distributions are introduced on the image side, leading to biased co-activation. Unfortunately, this bias tends to persist and even amplify during the training process, as weighting and gating distributions are often mutually dependent.

In Appendix C, we empirically measure the dependency of lexical retriever on MLM by initializing its projection layer within the DST head. The results demonstrate that once this is done, our model struggles to converge and becomes non-functional. Below we delve into a comprehensive exploration of these challenges and propose effective solutions to address them.

**(a) Activation amount.** Previous lexical retrievers utilize ReLU activation on $V(x)$ to introduce sparsity. While the $V(x)$ on image side is randomly initialized, the ReLU based gating produce uncontrollable amount of activation which will bias the learning. Specifically, an excessive number of activations could lead to co-activation on irrelevant dimensions while inadequate activation results in less effective learning, ultimately leading to suboptimal learning outcomes.

**elu1p activation with top-k sparsification.** Instead of ReLU, we adopt a combination of $elu1p$ activation with top-$k$ sparsification to ensure a deterministic number of activations. Moreover, during training, we fully activate $V_p(p)$ while sparsify $V_q(q)$, establishing a lower bound on the amount of co-activations for effective learning. This design also allows for a flexible trade-off between effectiveness and efficiency downstream by adjusting the activation amount.

**(b) Bias from co-activated dimensions.** Lexical retrievers have the capability to expand their semantic understanding by activating dimensions beyond the tokens presented in the input. However, this expansion can occasionally lead to unintended co-activations occurring in irrelevant dimensions, potentially introducing bias into the learning process. We refer to this situation as $\alpha_i(q, p) = 1$ while the $i$-th token in $\mathbb{V}$ fails to reflect the relevance between $q$ and $p$. This issue becomes more pronounced when combined with the inherent randomness introduced by $V_p(p)$. Once this phe-

nomenon takes place, rectification becomes challenging, and the bias often amplifies as weighting and gating distributions are often mutually dependent.

**Nonparametric entry.** To mitigate challenge (b), we introduce a nonparametric entry in loss computation. This entry uses bag-of-word vectors for queries, denoted as $sim(q, p) = \text{bow}(q) \cdot V_p(p) = \sum_{i \in T(q)} V_p(p)[i]$, where $T(q)$ is the set of tokens within $q$. This loss term compels $V_p(p)$ to concentrate on the tokens in $q$ without expansion, thereby preventing the irrelevant dimensions from possessing excessively large values.

**(c) Bias from inactive dimensions.** Another challenge arises from dimensions that remain inactive throughout the training process. This situation commonly arises with infrequent tokens and cross-lingual tokens. On the image side, these dimensions might exhibit substantial values during random initialization, and persist due to the lack of direct supervision. Let $T(B)$ denote the dimensions activated by $G_q(q)$ for all $q$ in batch $B$, the supervision only cover dimensions within $T(B) \subseteq \mathbb{V}$ while the remaining dimensions in $\mathbb{V} \backslash T(B)$ lack direct supervision as they do not explicitly contribute to the contrastive objective. Although some level of control is exerted through the normalization on $V(x)$, empirical evidence suggests that this is not sufficiently effective.

**Contrastive (CTS) mask.** We propose the design of contrastive mask to alleviate issue (c). Specifically, the contrastive mask distributes those neglected dimensions to the in-batch instances. For each $q_j$ from the batch $B$, we enforce $G_q(q_j)$ to activate $\frac{|\mathbb{V} \backslash T(B)|}{|B|}$ dimensions from the set $\mathbb{V} \backslash T(B)$, while concurrently deactivating them in $G_p(p_j)$. This will involve these dimensions into the computation of similarity of negative pairs $sim(q_j, p)$ for all $p \neq p_j^+$ while cancel their contribution to the positive pairs $sim(q_j, p_j^+)$. Through this approach, VDR introduces dimension-wise supervision across the entire $\mathbb{V}$ space, thereby facilitating a stable and reliable disentanglement learning process.

# 4 Experimental Setup

We conduct two groups of experiments for text-to-text and cross-modal retrieval scenarios, referred to as $\text{VDR}_{\text{t2t}}$ and $\text{VDR}_{\text{cm}}$, respectively.

## 4.1 Datasets

**Text-to-text.** we train $\text{VDR}_{\text{t2t}}$ on MS MARCO passage ranking dataset (Bajaj et al., 2016) which comprises approximately 8.8 million passages and around 500 thousand queries. We conduct zero-shot evaluations on 12 datasets from the BEIR benchmark (Thakur et al., 2021), which are widely used across previous papers.

**Cross-modal.** we utilize the mid-scale YFCC15M dataset introduced by DeCLIP (Cui et al., 2022), containing 15 million image-caption pairs for training. Our evaluation spans ImageNet, COCO Captions (Chen et al., 2015), and Flickr30k (Plummer et al., 2015) datasets.

## 4.2 Implementation Details

Our experimental settings and training configuration follow DPR under text-to-text scenarios and CLIP under cross-modal scenarios. We use AdamW optimizer (Loshchilov & Hutter, 2018) with a learning rate that linearly increases in the first epoch and then gradually decays. All of our models are trained on NVIDIA V100 GPUs with 32GB memory.

**Text-to-text.** We train $\text{VDR}_{\text{t2t}}$ for 20 epochs with a batch size of 256 and a learning rate of 2e-5. Each query is paired with one negative passage provided by the MS MARCO dataset during the training. We use tied embedding and do not incorporate contrastive masking in textual retrieval.

**Cross-modal.** We train $\text{VDR}_{\text{cm}}$ for 20 epochs with a batch size of 4096 and a learning rate of 2e-4. The input resolution of the image encoder is 224 × 224, and the max sequence length of the text encoder is 77. We initialize the learnable temperature parameter to 0.07, adopt the same prompt engineering and ensembling techniques as CLIP for ImageNet.

## 4.3 Evaluation

Our model offers two inference modes based on how the query $q$ is represented. **Parametric inference**, denoted as VDR, follows the conventional lexical retrieval pipeline as $E_q(q) = V_q(q) \cdot G_q(q)$. The optimal level of sparsity is determined through grid search, involving the variation of activation amount across diverse tasks. **Nonparametric inference**, referred to as $\text{VDR}^{\alpha}$, employs a normalized bag-of-words vector for query representation, i.e., $E_q(q) = bow(q)$. For both inference modes,

we adhere to the conventional pipeline to embed $p$ where $E_p(p) = V_p(p) \cdot G_p(p)$. For fair comparison, we set $k = d$ in all of our gating function, where $d = 768$ for text-to-text and $d = 512$ for cross-modal scenarios, which is the dimensions of the latent representation from the dense retriever baselines. We evaluate the disentanglement quality from two aspect:

**External aspect**: whether the disentangled representation of the natural language counterpart can identify and retrieve the corresponding data among the vast pool of candidates. We adhere conventional retrieval benchmark (§5) to assess this aspect.

**Internal aspect**: whether the dimensional values can effectively explain the input data. Given the absence of a definitive ground truth for this evaluation, we devise an indirect assessment method. In the retrieval benchmark, we gauge the retrieval accuracy of $VDR^\alpha$. For cross-modal retrieval, we additionally supplement our evaluation with case studies (§6.2) and human assessments (§6.2).

### 4.4 BASELINES

**Text-to-text.** We compare $VDR_{t2t}$ to several primary and advanced baselines. The primary baselines include BM25, SPLADE-max (Formal et al., 2021a), and DPR (Karpukhin et al., 2020). Notably, DPR and SPLADE have similar model sizes and training settings compared to our methods. The advanced retrieval baselines, include ANCE (Xiong et al., 2020), UnifieR (Shen et al., 2022b), Contriever (Izacard et al., 2021), SimLM (Wang et al., 2022a), MASTER (Zhou et al., 2022), RetroMAE (Liu & Shao, 2022), LexMAE (Shen et al., 2022a), and E5 (Wang et al., 2022b). These advanced baselines incorporate sophisticated techniques such as retrieval-oriented pre-training, specialized negative sampling, knowledge distillation, and access Wikipedia data during training. These additional techniques, while promising, often introduce considerable computational overhead and manual tuning efforts. More details can be found in Appendix J. Due to resource constraints, we will leave them for future work.

**Cross-modal.** We compare $VDR_{cm}$ primarily against CLIP (Radford et al., 2021), CLIP-BERT, and with advanced baselines SLIP (Mu et al., 2022), FILIP (Yao et al., 2021), DeCLIP (Li et al., 2021), and ProtoCLIP (Chen et al., 2022). Specifically, SLIP, DeCLIP, and ProtoCLIP incorporate within-modal supervision, which is known to be beneficial but computationally expensive (Andonian et al., 2022; Geng et al., 2023). FILIP leverages the finer-grained alignment between image patches and textual tokens. CLIP-BERT is a variant of CLIP that uses BERT as the text encoder, with a similar model size and training setting to us.

## 5 EXPERIMENTS

### 5.1 TEXT-TO-TEXT RETRIEVAL

| Model | BM25 | SPLADE | †DPR | †VDR$^\alpha_{t2t}$ | †VDR$_{t2t}$ | ANCE | UnifieR | Contriever | SimLM | MASTER | RetroMAE | LexMAE | E5$_{base}$ |
|---|---|---|---|---|---|---|---|---|---|---|---|---|---|
| Retrieval Pre-training | | | ✗ | | | | ✔ | ✔ | ✔ | ✔ | ✔ | ✔ | ✔ |
| Special Negatives | | | ✗ | | | ✔ | ✔ | | ✔ | ✔ | | ✔ | ✔ |
| Distillation | | | ✗ | | | | | | ✔ | ✔ | | ✔ | ✔ |
| Wikipedia Access | | | ✗ | | | | | ✔ | ✔ | ✔ | ✔ | | ✔ |
| ArguAna | 31.5 | 43.9 | 40.8 | **48.8** | 48.6 | 41.5 | 39.0 | 44.6 | 42.1 | 39.5 | 43.3 | 50.0 | 51.4 |
| Climate-FEVER | **21.3** | 19.9 | 16.2 | 18.1 | 17.6 | 19.8 | 17.5 | 23.7 | 16.3 | 21.5 | 23.2 | 21.9 | 15.4 |
| DBPedia | 31.3 | 36.6 | 30.4 | 37.6 | **39.0** | 28.1 | 40.6 | 41.3 | 34.5 | 39.9 | 39.0 | 42.4 | 41.0 |
| FEVER | **75.3** | 73.0 | 63.8 | 74.8 | 74.0 | 66.9 | 69.6 | 75.8 | 65.7 | 69.2 | 77.4 | 80.0 | 58.2 |
| FiQA | 23.6 | 28.7 | 23.7 | **29.3** | 28.8 | 29.5 | 31.1 | 32.9 | 29.2 | 32.8 | 31.6 | 35.2 | 36.4 |
| HotpotQA | 60.3 | 63.6 | 45.2 | **68.4** | 65.5 | 45.6 | 66.1 | 63.8 | 58.1 | 58.9 | 63.5 | 71.6 | 62.2 |
| NFCorpus | 32.5 | 31.3 | 26.1 | 32.7 | **33.0** | 23.7 | 32.9 | 32.8 | 32.3 | 33.0 | 30.8 | 34.7 | 36.6 |
| NQ | 32.9 | 46.9 | 43.2 | 45.8 | **47.2** | 44.6 | 51.4 | 49.8 | 47.7 | 51.6 | 51.8 | 56.2 | 60.0 |
| SCIDOCS | **15.8** | 14.5 | 10.9 | 15.4 | 15.3 | 12.2 | 15.0 | 16.5 | 14.5 | 14.1 | 15.0 | 15.0 | 19.0 |
| SciFact | 66.5 | 62.8 | 47.4 | **67.6** | 67.3 | 50.7 | 68.6 | 67.7 | 58.8 | 63.7 | 65.3 | 71.7 | 73.1 |
| TREC-COVID | 65.6 | 67.3 | 60.1 | **69.0** | 67.8 | 65.4 | 71.5 | 59.6 | 63.7 | 62.0 | 77.2 | 76.3 | 79.6 |
| Touché-2020 | **36.7** | 20.1 | 22.1 | 27.7 | 29.8 | 28.4 | 30.2 | 23.0 | 29.2 | 32.0 | 23.7 | 29.0 | 28.3 |
| Avg. | 41.1 | 42.4 | 35.8 | **44.6** | 44.5 | 38.0 | 44.5 | 44.3 | 44.4 | 43.1 | 45.1 | 48.7 | 46.8 |
| Avg. (w/o NQ) | - | - | - | 44.5 | 44.3 | - | - | 43.8 | 40.4 | 42.4 | 44.5 | - | 45.6 |

Table 1: Text-to-text retrieval results on BEIR benchmark (NDCG@10). †: our implementation. **Bold**: the best among primary baselines.

**Overall performance.** Table 1 presents a effectiveness comparison of different approaches on the BEIR benchmark in a zero-shot scenario. Notably, $VDR_{t2t}$ outperforms both DPR and SPLADE by a significant margin, while maintaining nearly identical model sizes and training configurations. When compared to advanced baseline methods, $VDR_{t2t}$ consistently outperforms the majority of them, without relying on sophisticated and computationally expensive techniques. It's worth noting that these techniques are orthogonal to the retrieval design and can be seamlessly integrated with our model to further enhance its effectiveness. This underscores the effectiveness of our proposed architectural design in text-to-text retrieval scenarios, even though it was initially designed to address challenges in cross-modal situations. The robust zero-shot performance exhibited by $VDR_{t2t}$

highlights its resilience and versatility, making it a strong foundation with the potential to benefit a wide range of applications.

**Remarkable effectiveness and efficiency of nonparametric inference.** It is worth noting that $VDR_{t2t}^{\alpha}$, which use binary bag-of-words query representations, surpasses the majority of baselines. This observation provides two essential insights. Firstly, our model excels in disentangling textual data, allowing the binary query representation to match the representation of its target from a vast pool of candidates, thereby delivering promising retriever accuracy. Secondly, its success underscores the nature of the current retrieval tasks, where term-based expansion and overlap play pivotal roles. It is worth highlighting that $VDR_{t2t}^{\alpha}$ does not need any neural network forwarding at inference time, thus significantly improving retrieval speed. In §6.3, we show that $VDR_{t2t}^{\alpha}$ can achieve over 10x efficiency compared to dense retrieval, making it an optimal choice under low-resource settings.

## 5.2 CROSS-MODAL RETRIEVAL

| Model | ImageNet | | MSCOCO | | | | | | | Flickr30k | | | | | | |
| | | | image-to-text | | | text-to-image | | | | image-to-text | | | text-to-image | | | |
| | Top1 | Top5 | R@1 | R@5 | R@10 | R@1 | R@5 | R@10 | R-mean | R@1 | R@5 | R@10 | R@1 | R@5 | R@10 | R-mean |
| CLIP | 32.8[†] | 57.4[†] | 20.8 | 43.9 | 55.7 | 13.0 | 31.7 | 42.7 | 32.6 | 34.9 | 63.9 | 75.9 | 23.4 | 47.2 | 58.9 | 50.7 |
| [†]CLIP-BERT | 32.4 | 56.1 | 23.9 | 47.8 | 60.3 | 13.6 | 33.8 | 45.1 | 37.4 | 44.1 | 71.2 | 80.7 | 27.8 | 54.7 | 65.9 | 57.4 |
| [†]VDR_cm | **38.7** | **63.6** | 30.9 | 54.5 | 65.4 | 17.4 | 38.1 | 49.7 | 42.7 | 51.0 | 79.3 | 86.7 | 32.4 | 60.1 | 70.7 | 63.4 |
| [†]VDR_cm^np | - | - | - | - | - | 11.8 | 28.6 | 38.6 | - | - | - | - | 21.1 | 42.3 | 52.8 | - |
| SLIP | 33.6[†] | 58.6[†] | 27.7 | 52.6 | 63.9 | 18.2 | 39.2 | 51.0 | 42.1 | 47.8 | 76.5 | 85.9 | 32.3 | 58.7 | 68.8 | 61.7 |
| [†]FILIP | 39.1 | 64.4 | 21.6 | 46.7 | 59.0 | 13.7 | 31.7 | 41.6 | 35.7 | 46.3 | 74.4 | 83.2 | 30.7 | 58.2 | 68.6 | 60.2 |
| ProtoCLIP | 32.0 | - | 30.2 | 55.1 | 66.5 | 16.9 | 37.9 | 49.4 | 42.7 | - | - | - | - | - | - | - |
| [†]DeCLIP | 43.2 | 69.4 | 25.3 | 51.2 | 63.4 | 16.6 | 35.2 | 45.4 | 39.5 | 51.3 | 80.7 | 88.5 | 35.5 | 63.0 | 73.0 | **65.3** |

Table 2: Cross-modal results on ImageNet, MS COCO, and Flickr30k. †: our implementations.

**Overall performance.** Table 2 presents the performance of cross-modal retrievers on the ImageNet, MS COCO, and Flickr30k datasets. The results demonstrate the effectiveness of our approach $VDR_{cm}$ across various benchmarks. Notably, $VDR_{cm}$ consistently surpasses the primary baseline models, CLIP and CLIP-BERT, following the same training conditions. Moreover, $VDR_{cm}$ exhibits superior performance compared to most advanced baselines. It is important to highlight that many of these advanced baselines rely on multi-vector representations or intra-modal objectives, which come at the expense of considerable computational demands during both training and inference stages.

In cross-modal scenarios, the performance of nonparametric inference $VDR_{cm}^{\alpha}$ is somewhat limited. We attribute this to the inherent nature of images, which often contain a wealth of information not explicitly presented in their captions. As a result, relying solely on the tokens present in queries for matching proves to be challenging, and the incorporation of expansion becomes essential.

## 6 ANALYSIS

### 6.1 ABLATION STUDIES

We have conducted extensive ablation studies within cross-modal scenarios to gain a deeper understanding of the individual components of our model. In order to mitigate computational expenses, we opted to train models for 5 epochs using a learning rate of 5e-4. We then evaluated the effects of removing specific components from the model. The results in Table 3

| | ImageNet | MS COCO | | Flickr30k | |
| | | TR | IR | TR | IR |
| VDR_cm | 29.3 | 25.4 | 13.7 | 43.2 | 26.9 |
| − CTS mask | 25.2 | 21.1 | 11.9 | 37.0 | 22.4 |
| − NP entry | 26.4 | 24.4 | 14.1 | 42.6 | 26.2 |
| − max pooling | 27.8 | 22.2 | 12.2 | 37.4 | 23.7 |

Figure 3: Ablation studies of different components of $VDR_{cm}$.

show that the removal of any of these components lead to a decline in performance across all three datasets. This highlights the positive impact of contrastive mask, nonparametric (NP) entry, and max pooling to $VDR_{cm}$.

### 6.2 INTERNAL INSPECTION OF DISENTANGLED REPRESENTATIONS

**Image disentanglement.** In Figure 4 (a,b), we conduct analysis of internal dimensional values within disentangled representations. The font size within the wordcloud indicates the dimensional values of this token in the weighting distribution $V(x)$. From the left, we observe that the vocabulary distribution remarkably aligns with the visual features of the image itself. On the right, red bounding boxes highlight specific patch groups for disentanglement. In this process, we apply max pooling only to the representations of these selected patches within the DST head. VDR effectively achieves disentanglement among different patch groups within the same image. This is particularly evident

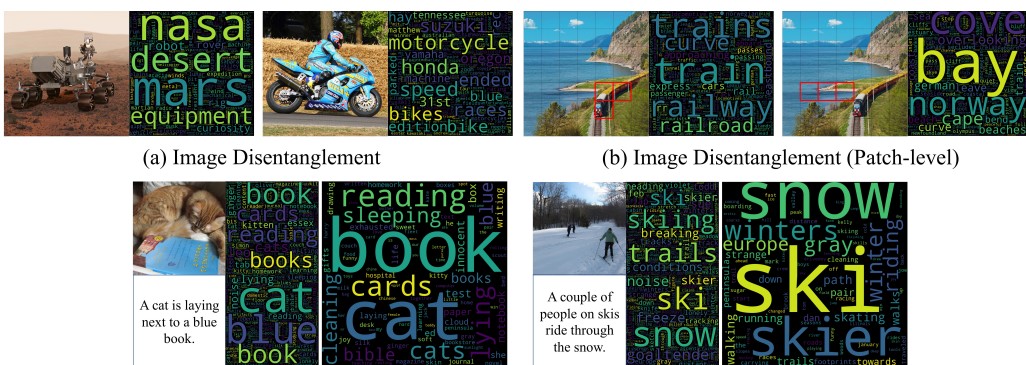

(a) Image Disentanglement         (b) Image Disentanglement (Patch-level)

(c) Retrieval Reasoning

Figure 4: Different approaches for internal inspection on disentangled representations.

in its ability to precisely associate separate visual concepts, such as "train" and "bay" with the respective patches in the image. Overall, these results vividly demonstrate that the disentangled representation generated by VDR exhibits rational dimensional values that efficiently explain the input data.

**Retrieval reasoning.** Figure 4 (c) shows the disentangled representation of the image $E_p(p)$, caption $E_q(q)$, and the element-wise product of both $E_q(q) \odot E_p(p)$, which provide insight into the retrieval process. In the example on the left, the dimensional values within the image representation prominently capture objects present in the image, such as "book" and "cat". On the other hand, the text representation emphasizes contextually relevant concepts like "reading" and "sleeping". By analyzing on $E_q(q) \odot E_p(p)$, VDR is able to quantify the individual contributions of each token to the retrieval process. This highlights a notable achievement of VDR in enhancing the interpretability of the retriever, enabling a clearer understanding of why certain objects are retrieved.

**Human evaluation.** We conducted human evaluations to compare $VDR_{cm}$ to the SOTA captioning model BLIP Li et al. (2022a) in terms of explainability. Details can be found in Appendix H. The results indicate that $VDR_{cm}$ achieves a satisfactory interpretation rate of 92%, outperforming BLIP, which achieves 85%. Notably, participants express a preference for our approach over BLIP in 48% of the cases, underscoring that our method effectively elucidates input data and matches the explainability of the leading captioning model.

### 6.3 RETRIEVAL EFFICIENCY

We perform retrieval using 1k queries with a corpus consisting of 100k data points. We employed inverted indexes for sparse retrieval. The detailed experimental setup is provided in Appendix G. We show retrieval effectiveness-efficiency of VDR with different inference modes in Figure G. The x-axis is retrieval latency per query, and the y-axis is the performance. Among the methods evaluated, the nonparametric inference $VDR^{\alpha}$ proved to be the most efficient, significantly outperforming parametric inference VDR. For VDR, fewer activations $k$ results in more sparse representations, which can enhance retrieval efficiency. When $k$ is less than 128, the efficiency of VDR is comparable to that of dense model.

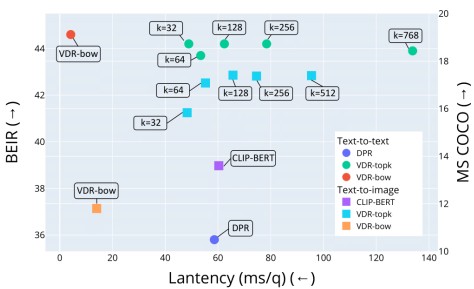

Figure 5: Effectiveness-efficiency comparisons of different retrievers.

### 7 CONCLUSIONS

In this work, we propose VDR, a simple but effective retrieval-based disentanglement framework that leverages natural language as a form of supervision. Our approach demonstrates that naturally occurring linguistic counterparts of data can effectively encourage the disentanglement on a vocabulary space. Extensive experiments and analysis show that VDR not only yields rational disentangled representations but also enhances the effectiveness, efficiency, and robustness of the retrieval system.

ACKNOWLEDGMENTS

We express our sincere gratitude to the anonymous reviewers for their invaluable comments and suggestions. Additionally, we extend our thanks to Lu Hou, Lewei Yao, and Haokun Lin from Noah's Ark Lab for their insightful discussions on multi-modal retrieval.

Lei Chen's work is partially supported by National Science Foundation of China (NSFC) under Grant No. U22B2060, National Key Research and Development Program of China Grant No. 2023YFF0725100, the Hong Kong RGC GRF Project 16213620, RIF Project R6020-19, AOE Project AoE/E-603/18, Theme-based project TRS T41-603/20R, CRF Project C2004-21G, China NSFC No. 61729201, Guangdong Basic and Applied Basic Research Foundation 2019B151530001, Hong Kong ITC ITF grants MHX/078/21 and PRP/004/22FX, Microsoft Research Asia Collaborative Research Grant and HKUST-Webank joint research lab grants.

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

## A  HYPERPARAMETERS

Our hyperparameter selection follows DPR for the text-to-text retrieval while CLIP for the cross-modal retrieval. Details can be found in Table 3 below.

|                          | text-to-text | cross-modal     |
|--------------------------|--------------|-----------------|
| Batch Size               | 256          | 4096            |
| Epoch                    | 20           | 20              |
| Learning Rate            | 2e-5         | 2e-4            |
| Warmup Epoch             | 1            | 1               |
| LR Decay                 | Linear       | CosineAnnealing |
| Normalization            | None         | L2-norm         |
| Temperature              | None         | 0.07            |
| Hard Negative            | 1            | 0               |
| Max Activation Num       | 768          | 512             |
| Max Seq Length (Q/P)     | 256/256      | 77/49           |
| Transformer Width (Q/P)  | 768/768      | 768/768         |

Table 3: Hyperparameters for training VDR.

## B  REPRODUCTION COMPARISONS

In Figure 6 and Table 4, we present our reproductions of DPR and CLIP, accompanied by results from other pertinent research papers. This facilitates valid reproduction and fair comparison. Notably, our study consistently showcases the highest levels of performance in relation to these foundational baselines.

Figure 6 showcases our replicated DPR model, which outperforms the versions reported in other studies. Therefore, we present our replicated baselines in main paper.

In Table 4, it's noteworthy that UniCLIP (Lee et al., 2022), having undergone pre-training on YFCC15M with a similar configuration, demonstrates superior outcomes. As a result, we have chosen to adopt their outcomes for the cross-modal retrieval aspect, with the exception of ImageNet where we have opted to utilize our own replicated scores.

| Model          | DPR  | DPR$^{\dagger}$ |
|----------------|------|------|
| MS MARCO       | 17.7 | 31.7 |
| ArguAna        | 17.5 | **40.8** |
| Climate-FEVER  | 14.8 | **16.2** |
| DBPedia        | 26.3 | **30.4** |
| FEVER          | 56.2 | **63.8** |
| FiQA           | 11.2 | **23.7** |
| HotpotQA       | 39.1 | **45.2** |
| NFCorpus       | 18.9 | **26.1** |
| NQ             | **47.4** | 43.2 |
| SCIDOCS        | 7.7  | **10.9** |
| SciFact        | 31.8 | **47.4** |
| TREC-COVID     | 33.2 | **60.1** |
| Touché-2020    | 13.1 | **22.1** |
| Avg.           | 26.4 | **35.8** |
| Best on        | 1    | 11   |

Figure 6: Reproduction of DPR from different sources. $\dagger$: ours.

| | ImageNet | | MSCOCO | | | | | | Flickr30k | | | | | |
|---|---|---|---|---|---|---|---|---|---|---|---|---|---|---|
| | | | image-to-text | | | text-to-image | | | image-to-text | | | text-to-image | | |
| | Top1 | Top5 | R@1 | R@5 | R@10 | R@1 | R@5 | R@10 | R@1 | R@5 | R@10 | R@1 | R@5 | R@10 |
| | | | | | Our re-implementation based on DECLIP's checkpoints | | | | | | | | | |
| CLIP$^{\dagger}$ | 32.80 | 57.35 | 16.94 | 39.50 | 50.94 | 10.75 | 26.19 | 35.24 | 34.70 | 65.00 | 74.10 | 23.60 | 46.90 | 58.76 |
| SLIP$^{\dagger}$ | 33.57 | 58.60 | 17.94 | 40.42 | 51.82 | 11.22 | 26.48 | 35.36 | 35.60 | 65.60 | 77.30 | 23.40 | 47.32 | 57.96 |
| FILIP$^{\dagger}$ | 39.16 | 64.35 | 21.64 | 46.66 | 59.00 | 13.72 | 31.72 | 41.60 | 46.30 | 74.40 | 83.20 | 30.66 | 58.18 | 68.56 |
| DeCLIP$^{\dagger}$ | 43.24 | 69.40 | 25.34 | 51.20 | 63.44 | 16.59 | 35.24 | 45.41 | 51.30 | 80.70 | 88.50 | 35.50 | 63.04 | 73.02 |
| | | | | | Results reported by UniCLIP | | | | | | | | | |
| CLIP | 31.3 | - | 20.8 | 43.9 | 55.7 | 13.0 | 31.7 | 42.7 | 34.9 | 63.9 | 75.9 | 23.4 | 47.2 | 58.9 |
| SLIP | 38.3 | - | 27.7 | 52.6 | 63.9 | 18.2 | 39.2 | 51.0 | 47.8 | 76.5 | 85.9 | 32.3 | 58.7 | 68.8 |
| DeCLIP | 41.2 | - | 28.3 | 53.2 | 64.5 | 18.4 | 39.6 | 51.4 | 51.4 | 80.2 | 88.9 | 34.3 | 60.3 | 70.7 |

Table 4: Reproduction of cross-modal retrieval on ImageNet, MS COCO, and Flickr30k from different sources.

## C    RELIANCE ON MASKED LANGUAGE MODEL

In this section, we empirically validate the reliance of lexical retriever on the pre-trained masked language models (MLM).

We adhere to the same training pipeline of our approach, while only initializing the linear projection within the DST head of the $p$ encoder and training it from scratch. This configuration is denoted as $VDR_{proj}$. Additionally, we incorporate BERT-based models as a lexical retrieval baseline, without undergoing further fine-tuning, denoted as $BERT_{lex}$. We present the training result below.

| Model | Epoch | NDCG10@BEIR | MRR10@MARCO |
|---|---|---|---|
| $BERT_{lex}$ | 0 | 20.1 | 28.9 |
| VDR | 1 | 38.9 | 28.4 |
| VDR | 2 | 42.3 | 30.8 |
| VDR | 3 | 42.9 | 31.7 |
| VDR | 4 | 43.4 | 32.4 |
| VDR | 5 | 43.7 | 32.8 |
| $VDR_{proj}$ | 5 | 0.2 | 0 |

Table 5: Different setup of lexical retrievers trained in the text-to-text retrieval scenarios.

Our experimental findings show that when we employ the pre-trained MLM projection, which inherently offers a rational weighting distribution from the outset, VDR reliably improve the effectiveness and achieve best results within 5 training epochs. Conversely, when starting from scratch with the projection layer on $p$ side, even with substantial training efforts, the $VDR_{proj}$ setup encounters challenges in attaining effective convergence. This obstacle compromises the final outcomes and makes it even fall behind the performance of the untrained baseline, $BERT_{lex}$. These findings support and validate the insights presented in Section 3.3.

Moreover, our observations and experiments in cross-modal retrieval suggest that achieving an effective transition from a scratch-initialized distribution to a rational one necessitates a substantial amount of training data, a large batch size, and the inclusion of the contrasting mask.

## D    IMPACT OF NONPARAMETRIC ENTRY

We emphasize the essential role of incorporating the nonparametric entry during training to achieve disentanglement in our model. Without it, our model tended to assign excessive values to overly common or rare tokens. We conjecture this issue arises from the interdependence between the gating and weighting functions, which amplifies biases rather than mitigating them.

To validate this hypothesis, we examine the embeddings produced by our model with and without the nonparametric entry. In Figure 7, we label our model in cross-modal setting with nonparametric entry as VDR (w/ BoW), without it as VDR (w/o BoW), and a BERT-based model as BERT-text. We take these encoders to embed text and images from the MS COCO test set into lexical representations, calculating average values for each token within these representations. We then visualize the top 100 tokens using word clouds and the distributions of their values using box plots. Our observations reveal that image representations from VDR (w/o BoW) have sharper distributions, characterized

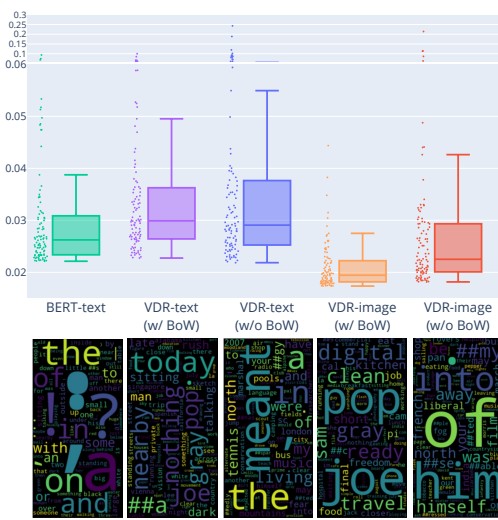

Figure 7: Box plot (top) and word cloud (bottom) of the vocabulary distributions on MS COCO.

by higher upper bounds and mean values in L2 norm space. However, the top 100 tokens in word cloud of VDR (w/o BoW) lacks meaningfulness and fails to convey distinct information. This implies that the omission of the nonparametric input in VDR amplifies biases, causing specific meaningless tokens to be assigned excessive values, thereby consistently dominating the matching outcome.

We term the above phenomenon as "disentanglement laziness". Simply put, the learning process avoids the "hard work" of properly disentangling and instead takes the easy route for optimization, degrading to a entangled fashion. In bi-encoder architecture, we have noticed that relying solely on parametric components causes the model to consistently assign high values to tokens that are either frequently or never encountered, resulting in entangled learning within a subset of the vocabulary space. In doing so, the model seemingly "escapes" from the rigorous work of disentanglement, reducing the problem into an optimization within an entangled representation space. Interestingly, this phenomenon is not exclusive to the research of disentangled representation learning. It is also observed in other research, like the Mixture of Experts (MoE), where it is referred to as "load imbalance". This term alludes to the model's tendency to consistently favor certain experts, thereby causing an unequal distribution of learning and optimization channels. In addressing the observed issue in our experiments, we enhance the disentanglement process by integrating the nonparametric entry, which provides stable and straightforward supervision of the data, independent of any influences from the entangled parametric model.

## E    SPARSITY V.S. EFFECTIVENESS

### E.1    AMOUNTS OF ACTIVATION

We present the effectiveness of VDR with different amounts of activation $k$ in Table 6 and Table 7.

| Model | Word Length | | VDR$^\alpha$ | VDR | | | | | |
|---|---|---|---|---|---|---|---|---|---|
| | Query | Doc | | 0* | 32 | 64 | 128 | 256 | 768 |
| MS MARCO | - | - | 33.8 | 33.0 | 34.1 | 34.4 | **34.5** | 34.4 | 34.3 |
| ArguAna | 193 | 167 | **48.8** | 48.6 | 27.3 | 41.7 | 47.0 | 47.2 | 46.5 |
| Climate-FEVER | 20 | 85 | **18.1** | 17.2 | 17.1 | 17.6 | 17.2 | 17.2 | 16.9 |
| DBPedia | 5 | 50 | 37.6 | 35.1 | 38.0 | 38.6 | **39.0** | 38.8 | 38.9 |
| FEVER | 8 | 85 | **74.8** | 73.7 | 74.0 | 73.9 | 73.9 | 73.9 | 73.9 |
| FiQA | 11 | 132 | **29.3** | 28.1 | 28.2 | 28.8 | 28.8 | 28.6 | 28.4 |
| HotpotQA | 18 | 46 | **68.4** | 64.4 | 65.0 | 65.5 | 65.5 | 65.4 | 65.0 |
| NFCorpus | 3 | 232 | 32.7 | 32.5 | **33.0** | 32.9 | 32.9 | 32.8 | 32.5 |
| NQ | 9 | 79 | 45.8 | 44.6 | 45.8 | 46.4 | 46.9 | 47.0 | **47.2** |
| SCIDOCS | 9 | 176 | **15.4** | 14.8 | 14.8 | 15.0 | 15.1 | 15.2 | 15.3 |
| SciFact | 12 | 214 | **67.6** | 67.3 | 66.8 | 67.2 | 67.1 | 67.3 | 66.6 |
| TREC-COVID | 11 | 161 | **69.0** | 66.5 | 67.3 | 67.8 | 67.6 | 67.3 | 66.2 |
| Touché-2020 | 7 | 292 | 27.7 | 29.1 | 29.0 | 29.4 | 29.5 | **29.8** | 29.4 |
| average | - | - | 44.6 | 43.5 | 42.2 | 43.7 | 44.2 | 44.2 | 43.9 |

Table 6: Effectiveness of VDR with varying activation amounts $k$. **Bold** denotes the overall best result and underline denotes the best query sparsity for VDR.

| VDR K | ImageNet | | MSCOCO | | | | | | Flickr30k | | | | | |
|---|---|---|---|---|---|---|---|---|---|---|---|---|---|---|
| | | | image-to-text | | | text-to-image | | | image-to-text | | | text-to-image | | |
| | Top1 | Top5 | R@1 | R@5 | R@10 | R@1 | R@5 | R@10 | R@1 | R@5 | R@10 | R@1 | R@5 | R@10 |
| 32 | 34.8 | 56.7 | 18.9 | 38.9 | 49.3 | 15.8 | 35.7 | 46.8 | 34.9 | 59.2 | 70.3 | 31.7 | 58.0 | 68.5 |
| 64 | 36.6 | 59.7 | 23.3 | 45.5 | 56.1 | 17.1 | 37.3 | 48.6 | 40.9 | 67.9 | 78.2 | 33.3 | 59.8 | 71.3 |
| 128 | 38.1 | 62.0 | 27.0 | 49.7 | 61.1 | **17.4** | **38.1** | 49.4 | 44.9 | 73.9 | 83.4 | **33.3** | 60.0 | **71.4** |
| 256 | 38.5 | 63.3 | 29.5 | 53.9 | 64.4 | **17.4** | **38.1** | 49.4 | 49.9 | 77.2 | 85.6 | 32.9 | 60.0 | 71.2 |
| 512 | **38.7** | **63.6** | 30.9 | 54.5 | 65.4 | **17.4** | **38.1** | 49.7 | **51.0** | **79.3** | **86.7** | 32.4 | **60.1** | 70.7 |

Table 7: Effectiveness of VDR with varying activation amounts $k$. **Bold** denotes the best result.

In the text-to-text scenario, the results demonstrate that the effectiveness of VDR increases as $k$ increases, reaching a peak and then decreasing. This suggests that by properly selecting the number of activation units, VDR is able to achieve considerable improvement.

In the cross-modal scenario, the results demonstrate that the effectiveness of VDR increases consistently as $k$ increases in the majority of cases. This suggests that a higher number of activation units can lead to better performance in cross-modal scenarios.

### E.2 AMOUNT OF VOCABULARY SIZE

We explore the effects of expanding the vocabulary size by switching from standard BERT encoders to multilingual BERT (mBERT) encoders with a vocabulary size of 110k, which is 4 times larger than that of BERT encoders. Our aim is to evaluate if a larger vocabulary and the consequent increase in sparsity would adversely affect our representation learning approach.

|  | DPR(mBERT) | VDR(mBERT) | DPR(BERT) | VDR(BERT) |
|---|---|---|---|---|
| ArguAna | 37.2 | 46.7 | 40.8 | 48.6 |
| Climate-FEVER | 14.6 | 14.7 | 16.2 | 17.6 |
| DBPedia | 31.3 | 36.4 | 30.4 | 39.0 |
| FEVER | 62.7 | 69.7 | 63.8 | 74.0 |
| FiQA | 21.6 | 27.5 | 23.7 | 28.8 |
| HotpotQA | 45.6 | 63.1 | 45.2 | 65.5 |
| NFCorpus | 25.2 | 33.3 | 26.1 | 33.0 |
| NQ | 43.2 | 43.8 | 43.2 | 47.2 |
| SCIDOCs | 10.7 | 14.3 | 10.9 | 15.3 |
| SciFact | 48.2 | 66.7 | 47.4 | 67.3 |
| TREC-COVID | 57.3 | 66.7 | 60.1 | 67.8 |
| Touché-2020 | 21.9 | 27.7 | 22.1 | 29.8 |
| Avg | 34.9 | 42.6 | 35.8 | 44.5 |

Table 8: Effectiveness of DPR and VDR using BERT-based and multilingual BERT-based encoders.

Table 8 illustrates the results with increased vocabulary size. With the switch to mBERT-based encoders, the vocabulary size grows from 30k to 110k, consequently increasing the sparsity due to the activation of the same number of dimensions. Interestingly, there is a minor performance drop in VDR on the BEIR benchmark, from 44.5 to 42.6, when switching to mBERT encoders. We also observe that DPR exhibits a similar trend. Notably, VDR outperforms DPR by approximately 23% in both scenarios, suggesting that the performance drop may not be directly linked to the larger vocabulary size. Instead, it could arise from a mismatch between the multilingual encoder and the predominantly English downstream retrieval tasks. This finding aligns with our earlier analysis in Section 3.3, where we posited that pretrained masked language models provide a solid foundation for establishing effective gating distributions. This experiment indicates that increasing the vocabulary size or sparsity does not significantly hinder the learning process.

## F CASE STUDY

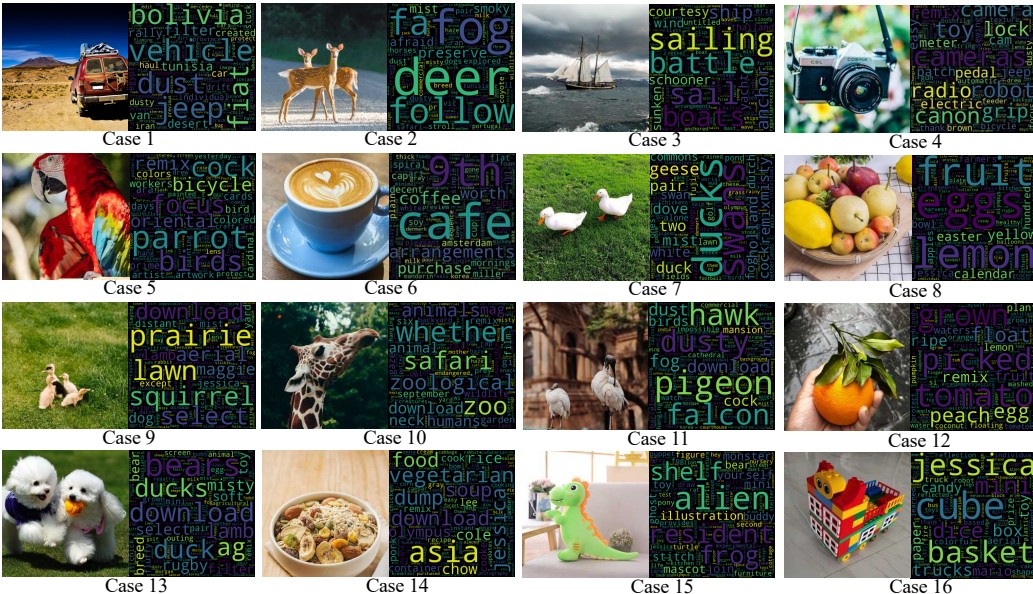

Figure 8: More case study on VDR disentanglement of image.

We provide additional case studies in Figure 8. Cases 1 through 8 represent successful cases as determined by our experts, while cases 9 through 16 illustrate instances where the image encoder did not perform as expected. For those good cases, we can observe that the main concepts present in the images are aptly represented in the word cloud. This indicates that the image encoder of VDR effectively captures the semantic meaning of these images, producing a reasonable and understandable representation within the disentangled vocabulary space. For the unsuccessful cases, we observed some cases of misidentification or misconception. After analysis, we identified two main reasons for these errors. First, there are cases where the encoder fails to correctly identify the object within the image because it resembles another object, thus skewing the results. For example, in cases 9 and 13, the encoder incorrectly identifies ducks as squirrels and dogs as bears, likely due to their similar appearances within the images. Secondly, certain images entail concepts associated with n-gram phrases, which is challenging for internal inspection or word cloud visualization. For instance, in case 10, the term "giraffe" is tokenized into three tokens: "gi", "##raf", and "##fe". While the first token, "gi", appears in the word cloud, the latter two are missing. Such n-gram concepts can be challenging to capture or infer through an internal inspection of the representation. This limitation can be traced back to the choice of tokenizer used prior to training.

## G   DETAILS IN EFFICIENCY MEASUREMENT

We perform retrieval using 1k text queries with a pre-embedded corpus consisting of 100k data points. We employed inverted indexes for sparse retrieval. The retrieval experiments are conducted on a single-threaded Linux machine with two 2.20 GHz Intel Xeon Gold 5220R CPUs. The batch size used in the experiments is one and the maximum sequence length for queries is 77. The MS MARCO and MS COCO datasets were utilized for text-to-text and text-to-image retrieval, respectively. The average query length for text-to-text retrieval was 6.8 and for text-to-image retrieval was 11.6. The effectiveness of the retrieval methods was evaluated using the average NDCG@10 scores on the BEIR metric for text-to-text retrieval and the Recall@1 metric for text-to-image retrieval on the MS COCO dataset.

## H   DETAILS OF HUMAN EVALUATION

This section outlines our evaluation approach to comparing the VDR with the SOTA captioning model BLIP. For VDR, we encoded images into disentangled representations. Human evaluators then selected the most understandable tokens from the top-5 dimensions, without access to the original image. In the case of BLIP, evaluators chose up to 5 tokens from BLIP-generated captions that they felt best captured the essence of the caption, again without seeing the corresponding images. We also included evaluations of the full captions generated by BLIP for comparison.

To assess the effectiveness of these methods, we recruited an additional group of 10 participants from universities with diverse background, each evaluating 20 images. They were tasked with determining (1) the effectiveness of the token sets in capturing key concepts of the images and (2) which token set (VDR or BLIP) provided a more accurate description of the image. To mitigate potential position bias, we randomized the order of the top-5 manually selected tokens from VDR and varied the presentation order of the two methods for each participant. All participants were presented with the same set of image samples in both phases of the evaluation. This ensured they had adequate familiarity and information for comparison in the subsequent phase.

Our findings indicated a high level of satisfaction among participants with the tokens derived from VDR, with 92% expressing satisfaction. This satisfaction rate was comparatively higher than the 85% satisfaction for the top-5 tokens selected from BLIP captions. However, it's noteworthy that satisfaction decreased to 76% when evaluating full BLIP captions. We interpret this decline as a result of two factors: firstly, BLIP captions are generally concise (ranging from 2 to 10 words), often making the top-5 tokens adequate for summarizing key concepts. Secondly, while full captions offer a comprehensive description, they sometimes include specific details (like location and quantity) that may not always be accurate, leading to a reduction in overall satisfaction. When comparing the top-5 tokens from both VDR and BLIP, participants showed a preference for our VDR approach in 48% of the evaluations. This demonstrates that our framework effectively captures and elucidates the essential elements of the input data, rivaling the explainability of the leading captioning model.

# I   ABLATION STUDY ON TEXT-TO-TEXT RETRIEVAL

We present an ablation study for text-to-text retrieval, aiming to understand the impact of different components on the performance of our text-to-text retrieval model. We evaluate various ablations to gain insights into the model's behavior. Specifically, we consider three ablations: the original implementation, "w/o elu1p" setting where we replace the elu1p activation with the relu activation, and "w/o bow" setting where we remove non-parametric entry loss during training.

| | $VDR_{t2t}$ | | | $VDR_{t2t}^{\alpha}$ | | |
|---|---|---|---|---|---|---|
| elu1p | ✔ | ✘ | ✔ | ✔ | ✘ | ✔ |
| bow | ✔ | ✔ | ✘ | ✔ | ✔ | ✘ |
| | NDCG@10 | | | | | |
| ArguAna | 48.6 | 45.5 | 43.5 | 48.8 | 47.9 | 41.4 |
| Climate-FEVER | 17.6 | 16.5 | 18.4 | 18.1 | 17.6 | 15.5 |
| DBPedia | 39.0 | 37.6 | 38.3 | 37.6 | 37.4 | 34.4 |
| FEVER | 74.0 | 72.2 | 73.7 | 74.8 | 72.6 | 70.7 |
| FiQA | 28.8 | 29.0 | 28.1 | 29.3 | 28.3 | 25.3 |
| HotpotQA | 65.5 | 63.8 | 64.5 | 68.4 | 68.4 | 63.2 |
| NFCorpus | 33.0 | 32.4 | 33.2 | 32.7 | 32.4 | 32.3 |
| NQ | 47.2 | 45.5 | 44.9 | 45.8 | 45.0 | 38.8 |
| SCIDOCs | 15.3 | 14.5 | 14.7 | 15.4 | 15.2 | 14.0 |
| SciFact | 67.3 | 65.6 | 65.7 | 67.6 | 67.1 | 66.0 |
| TREC-COVID | 67.8 | 64.8 | 68.0 | 69.0 | 66.7 | 60.7 |
| Touché-2020 | 29.8 | 28.8 | 29.4 | 27.7 | 27.2 | 21.4 |
| Avg | 44.5 | 43.0 | 43.5 | 44.6 | 43.8 | 40.3 |

Table 9: Ablation study for text-to-text retrieval on BEIR benchmark.

# J   FAIRNESS IN BASELINE DISTINCTION

The advanced baselines refer to baselines that employ sophisticated techniques known for significantly improving the performance number but orthogonal to the retrieval design. These methods often come with a significant increase in computational requirements and the need for meticulous tuning. Here, we delve into these techniques as applied in a text-to-text scenario. (1) Retrieval-oriented pre-training involves using large-scale pre-training tasks (Chang et al., 2020) tailored to improve the retriever's efficiency. (2) Specialized negative sampling is well known as crucial in contrastive learning and training retriever (Lin et al., 2021). (3) Knowledge distillation (Gou et al., 2021) is a process where knowledge from a larger model (usually cross-encoder) is transferred to a simpler one (bi-encoder). While this improves performance, it also increases complexity and computational demands. (4) Access to Wikipedia data during training (Ren et al., 2022) potentially benefit the performance on datasets constructed from Wikipedia, such as the NQ dataset.

We primarily compare our method, $VDR_{t2t}$, with DPR, and $VDR_{cm}$ with CLIP-BERT. These comparisons are apt because these methods have identical parameter counts and follow similar training pipelines. Both $VDR_{t2t}$ and DPR have 217 million parameters and were trained on 500k question-passage pairs using 8 V100 GPUs over one day. Similarly, both $VDR_{cm}$ and CLIP-BERT, having an equal number of 197 million of parameters, were trained on 15 million image-caption pairs for six days.

# K   SIGNIFICANCE TEST ON IMPROVEMENT

The results of significance tests comparing the performance of the VDR against baseline models with identical parameter counts and training pipelines are detailed below. These tests were designed to evaluate whether VDR offers a statistically significant improvement over the baseline models, with the null hypothesis positing equal performance between the compared models. For text-to-text retrieval, the results are summarized in Table 9. Out of 12 datasets, $VDR_{t2t}$ showed statistically significant improvements in 9 datasets, as indicated by the symbol △. Similarly, for cross-modal retrieval, as detailed in Table 10, VDRcm exhibited significant improvements across all tested settings when compared to the CLIP-BERT baseline. These results collectively indicate that VDR, in both text-to-text and cross-modal scenarios, offers substantial improvements over the baseline models.

| Model | MSCOCO | | | | | | Flickr30k | | | | | |
| | image-to-text | | | text-to-image | | | image-to-text | | | text-to-image | | |
| | R@1 | R@5 | R@10 | R@1 | R@5 | R@10 | R@1 | R@5 | R@10 | R@1 | R@5 | R@10 |
| CLIP-BERT | 23.9 | 47.8 | 60.3 | 13.6 | 33.8 | 45.1 | 44.1 | 71.2 | 80.7 | 27.8 | 54.7 | 65.9 |
| $VDR_{cm}$ | 30.9△ | 54.5△ | 65.4△ | 17.4△ | 38.1△ | 49.7△ | 51.0△ | 79.3△ | 86.7△ | 32.4△ | 60.1△ | 70.7△ |

Table 10: Statistical significance for differences with $VDR_{cm}$ and CLIP-BERT via a two-sided student-t test, △ indicates methods with significantly higher Recall with $p < 0.01$.

| | DPR | $VDR_{t2t}$ |
| --- | --- | --- |
| ArguAna | 40.8 | 48.8△ |
| Climate-FEVER | 16.2 | 18.1△ |
| DBPedia | 30.4 | 37.6△ |
| FEVER | 63.8 | 74.8△ |
| FiQA | 23.7 | 29.3△ |
| HotpotQA | 45.2 | 68.4 |
| NFCorpus | 26.1 | 32.7△ |
| NQ | 43.2 | 45.8△ |
| SCIDOCs | 10.9 | 15.4△ |
| SciFact | 47.4 | 67.6△ |
| TREC-COVID | 60.1 | 69.0 |
| Touché-2020 | 22.1 | 27.7 |

Figure 9: Statistical significance for differences with $VDR_{t2t}$ and DPR via a two-sided student-t test, △ indicates methods with significantly higher NDCG@10 with $p < 0.01$.

