# OpenReview forum: "Retrieval-based Disentangled Representation Learning with Natural Language Supervision"
_ICLR.cc/2024/Conference — ICLR 2024 spotlight_

### Official Review · Reviewer_LxNZ · 2023-11-02

**Soundness:** 4 excellent
**Presentation:** 4 excellent
**Contribution:** 3 good
**Rating:** 6
**Confidence:** 4

**Summary:**

The paper introduces an innovative approach called Vocabulary Disentangled Retrieval (VDR) for learning disentangled representations using natural language as a supervisory signal. VDR employs a bi-encoder architecture that is trained on data-text pairs, where a disentanglement head maps dense representations into a disentangled space. This space is structured such that each dimension correlates with a token from the BERT tokenizer's vocabulary, creating an interpretable, token-level representation of data.

The authors present two novel sparsification techniques to refine these disentangled representations: top-k sparsification and nonparametric sparsification using normalized binary bag-of-words. These methods are designed to maintain the interpretability of the representations while enhancing retrieval performance.

Extensive experiments across 15 benchmark datasets, encompassing text-to-text and text-to-image retrieval tasks, demonstrate that VDR significantly outperforms comparable bi-encoder retrievers and existing baselines.

By utilizing sparse embeddings and a vocabulary-defined space without complex or computationally intensive methods, the paper's proposed approach provides a more interpretable and efficient solution for embedding-based retrieval across various domains.

**Strengths:**

The approach provides interpretable representations that are robust for retrieval tasks, though not necessarily state-of-the-art. The quality of the work is reflected in its thorough experimental setup, with evaluations that cover both document and image retrieval tasks.

The clarity of the paper is another strength, as the paper is well-written and easy to understand. This clarity extends to the simplicity and ease of implementing the proposed retrievers, which also generalize to different applications such as text-to-text and text-to-image retrieval tasks.

In terms of significance, the model demonstrates a strong performance in zero-shot retrieval tasks. The VDR-bag of words, in particular, is competitive with the more complex and parametric VDR-k, offering better latency and making it appealing for low-resource applications. The encouraging results on BEIR benchmark and cross-modal benchmarks underscore the impact of the work, suggesting that the method could be further improved by integrating it with other orthogonal approaches.

**Weaknesses:**

In my opinion, the biggest concern with this paper is that its main innovation being the use of vocabulary as a basis for disentanglement, which is not convincingly demonstrated to enhance explainability or offers significant improvements.

The focus of the methodology and experiments appears to be on training effective retrievers using disentangled representation in the vocabulary space. However, this seems to diverge from the initial motivation, which emphasizes using natural language to guide disentanglement learning. This inconsistency complicates the narrative flow and understanding of the paper’s objectives.

In terms of empirical evaluation, while the model demonstrates improved performance in text-to-text and cross-modal retrieval tasks, there is a lack of analysis on non-retrieval tasks. This omission limits the understanding of how the disentanglement contributes to broader applications beyond retrieval.

**Questions:**

The authors use elu1p activation as a replacement for ReLU. Do you have experimental results that motivate this decision?

Do the authors have any intuition as to why their approach works better than DPR? It is not clear to me why dense representations would perform much worse than the disentangled representations based on the vocabulary.

---

> ### Author Response · Authors · 2023-11-15
> **Response to Reviewer LxNZ**
>
> Thank you for your detailed review and insightful feedback. We address your comments and questions below.
>
> **W1: Not convincingly demonstrated to enhance explainability.**
>
> We appreciate the feedback and recognize the critical importance of assessing the explainability. Indeed, evaluating the explainability of any system is an continuous challenge, which often necessitates human assessment. Especially in the disentanglement field, where a universally accepted evaluation methodology is yet to be established [1]. Despite these challenges, we are fully committed to demonstrating the explainability of our framework through a comprehensive approach that encompasses human evaluation (Section 6.2, Appendix H), real-world case studies (Section 6.2 and Appendix F), and retrieval benchmarking.
>
> In addition, it is worth noting that our framework's explainability can be partially assessed through non-parametric inference. This involves comparing the similarity between the BoW representations of queries and representations of their corresponding target data. The similarity score indicates how well the dimensional values within $p$ representation reflects the tokens presented in $q$. Although this method only utilizes half of the model's parameters and being stringent, VDR framework still shows effective performance in text-to-text and fair performance in cross-modal retrieval. This indirectly supports its explainability.
>
> **W2: Lack of analysis on non-retrieval tasks.**
>
> Thanks for the feedback! Our current research primarily focuses on retrieval benchmarks, where well-established datasets and comparable baselines allow for a robust evaluation to validate our approach. However, we acknowledge the importance of extending this research to a wider range of applications beyond retrieval. We consider this a foundational step towards their broader application and eagerly anticipate future advancements in this area.
>
> **Q1: The authors use elu1p activation as a replacement for ReLU. Do you have experimental results that motivate this decision?**
>
> We have conducted a comprehensive ablation study for text-to-text retrieval, detailed in Appendix I. The result demonstrates that using elu1p activation alongside top-k sparsification resulted in an average 3% improvement in NDCG@10 on the BEIR benchmark compared to using ReLU activation.
>
> | |VDR|VDR(w/o elu1p)|VDR_a|VDR_a(w/o elu1p)|
> |-|:-:|:-:|:-:|:-:|
> |ArguAna|48.6|45.5|48.8|47.9|
> |Climate-FEVER|17.6|16.5|18.1|17.6|
> |DBPedia|39.0|37.6|37.6|37.4|
> |FEVER|74.0|72.2|74.8|72.6|
> |FiQA|28.8|29.0|29.3|28.3|
> |HotpotQA|65.5|63.8|68.4|68.4|
> |NFCorpus|33.0|32.4|32.7|32.4|
> |NQ|47.2|45.5|45.8|45.0|
> |SCIDOCs|15.3|14.5|15.4|15.2|
> |SciFact|67.3|65.6|67.6|67.1|
> |TREC-COVID|67.8|64.8|69.0|66.7|
> |Touché-2020|29.8|28.8|27.7|27.2|
> |Avg|44.5|43.0|44.6|43.8|
>
> **Q2: Do the authors have any intuition as to why VDR works better than DPR?**
>
> The reviewer's question highlights an important aspect of our work! We share two key insights from representation learning perspective:
>
> 1. VDR capitalizes on the strengths of masked language models (MLM) more effectively than DPR.
>     - MLM pre-training projects $d$-dimensional latent representations to |V|-dimensional vocabulary space. These projected |V|-dimensional representations are optimized to predict masked tokens, leading to reasonable vocabulary distributions where high dimensional values reflect great contextual relevance to the corresponding tokens. VDR harnesses this by employing an elu1p activation and max-pooling over these |V|-dimensional representations, aligning closely with the nature of text matching tasks.
>     - DPR relies on the [CLS] token representation, which are originally optimized for next sentence prediction in MLM pre-training, creating a gap in suitability for information retrieval tasks.
>     - We provide empirical evidence supporting this in below table where VDR outperforms DPR significantly when using a BERT-based encoder without any fine-tuning.
>
> | |DPR|VDR|
> |-|-|-|
> |ArguAna|4.9|42.0|
> |Climate-FEVER|1.5|3.6|
> |DBPedia|0.5|7.6|
> |FEVER|1.0|7.1|
> |FiQA|0.0|4.3|
> |HotpotQA|0.2|21.4|
> |NFCorpus|1.6|7.1|
> |NQ|0.1|6.1|
> |SCIDOCs|0.3|4.6|
> |SciFact|2.2|31.8|
> |TREC-COVID|2.3|20.2|
> |Touché-2020|0.0|3.4|
> |Avg|1.2|13.3|
>
> 2. VDR utilizes a significantly larger representation space than DPR. This expanded space allows for a greater capacity to capture nuanced features in data. This concept aligns with the principles in mixture-of-experts (MoE) research [2]. Just as MoE increases network capacity through a larger number of selectively activated parameters, VDR expands its representational capability through increased dimensionality and introduces sparsity for supervision and fast downstream inference.
>
> [1] Carbonneau, Marc-André, et al. "Measuring disentanglement: A review of metrics."
>
> [2] Shazeer, Noam, et al. "Outrageously large neural networks: The sparsely-gated mixture-of-experts layer."

---

> ### Author Response · Authors · 2023-11-20
> **Gentle Reminder**
>
> Dear Reviewer LxNZ,
>
> We sincerely appreciate the time and effort you have dedicated to reviewing our paper. Your insightful questions and valuable feedback have greatly contributed to the enhancement of our research.
>
> In response to your valuable feedback, we have diligently worked on a rebuttal that addresses concerns you raised. As the discussion period is approaching its conclusion in two days, we would greatly appreciate your review of our rebuttal at your earliest convenience. Should there be any further points that require clarification or improvement, please know that we are fully committed to addressing them promptly.
>
> Thank you once again for your invaluable contribution to our work.
>
> Warm Regards,
>
> The Authors

---

### Official Review · Reviewer_LJaD · 2023-11-06

**Soundness:** 2 fair
**Presentation:** 2 fair
**Contribution:** 2 fair
**Rating:** 6
**Confidence:** 3

**Summary:**

This work introduces Vocabulary Disentangled Retrieval (VDR), a method that aims to take a retrieval-based approach to achieve disentangled representation learning. VDR supports parametric and non-parametric inference. The latter option does not require a neural network during inference, decreasing the computation speed, making it suitable for low-resource scenarios. The method is tested on text-to-text retrieval and cross-model retrieval tasks. The baselines are split into primary and advanced baselines, and VDR outperforms many of the primary baselines in terms of NDCG and Recall. A human evaluation on 20 images, and a case study with 6 images is presented to evaluate the disentanglement of VDR.

**Strengths:**

- The proposed method, VDR, supports nonparametric inference, which increases the inference efficiency. This makes this method an interesting option in lower resource settings (given budget for training).
- The authors address an important problem, and their method could be used to help users interpret the results of retrieval system.

**Weaknesses:**

- I am mostly concerned about the evaluation methods used, specifically:

(1) The retrieval baselines are split into primary baselines and advanced baselines. The authors position themselves in the primary group, arguing that their method does not need advanced techniques as in the advanced baseline group. I am not sure how 'advanced' is defined here, and thus whether this is a a fair distinction. This is important, as VDR outperforms the primary baselines, but does not necessarily outperform the advanced baselines.

(2) One of the most important aspects of VDR is the disentangled representation. This is evaluated with a small case study of only 6 images (and some more in the appendix) and a small scale human evaluation. Except for the small size of the data samples, it is also not clear how the images were selected. Are these cherry picked, or random? There is also little information about the setup of the human evaluation, making it difficult to judge these results. Given that this is such an important part of the paper, I would have expected a larger scale evaluation.

- The paper makes a couple of unsubstantiated claims, most notably:

(1) Page 2: "In practice, individuals employ natural language to interpret and distinguish objects." There is no source to back up this claim, and it is also not entirely clear what the authors mean by this, for example, babies can distinguish objects, before they can speak?

(2) The definition of Information Retrieval on page 3 is very narrow. It does not follow the broader definition given in Manning, 2009, which the authors cite. The other source, Zhao et al., 2022, is a survey paper on dense text retrieval methods, which is also only a tiny fraction of the entire field of Information Retrieval.

- Minor detail: the authors cite Kingma & Ba (Adam) for AdamW, instead of Loshchilov & Hutter.

**Questions:**

- The current work focusses on disentanglement, and I am wondering what the authors consider to be the difference between disentanglement and grounding in this case?

- The authors claim that VDR outperforms the baselines with a significant margin, can the authors give the results of a significance test that backs up this claim?

---

> ### Author Response · Authors · 2023-11-15
> **Response to Reviewer LJaD (Part 1/2)**
>
> Thank you for your detailed review and insightful feedback. We address your comments and questions below.
>
> **W1(1): How 'advanced' baseline is defined and whether this is a a fair distinction?**
>
> The "advanced baselines" refer to baselines that employ sophisticated techniques known for significantly improving the performance number but orthogonal to the retrieval design. These techniques often introduce substantial computational cost and tuning effort.
>
> While we have detailed these techniques in cross-modal scenario in Section 4.4, below we offer additional details about these techniques in text-to-text scenario:
> - Retrieval-oriented pre-training utilizes large-scale pre-training tasks`[1]` specifically designed to boost retriever.
> - Specialized negative sampling is well known as crucial in contrastive learning and training retriever.`[2]`
> - Knowledge distillation`[3]` is a process where knowledge from a larger model (usually cross-encoder) is transferred to a simpler one (bi-encoder). While this improves performance, it also increases complexity and computational demands.
> - Access to Wikipedia data during training potentially benefit the performance on datasets constructed from Wikipedia, such as the NQ dataset. `[4]`
>
> **W1(2): Issues related to evaluations.**
>
> - The examples for case study and human evaluation were randomly selected.
> - We provide more details about the setup and analysis of human evaluation in Appendix H.
> - In addition to case studies and human evaluations, our framework's explainability can be also demonstrated through a retrieval benchmark. This is significant for the following reasons. Our framework utilizes disentangled representations, which are notably sparse (< 2% activation). This sparse nature ensures that the similarity between query-target pairs ($q_i$ and $p_i$) is partially dependent on the number of dimensions they both activate. Achieving high performance in retrieval tasks require the representations of each pair ($q_i$, $p_i$) assgin large values to the dimensions that reflect their data characteristics in common,  while assigning lower values to dimensions representing other potential targets $p_j$ in the large corpus. Such a mechanism indirectly validates the effectiveness and reliability of our disentangled representations.
>
> **W2: A couple of unsubstantiated claims and citation error for AdamW.**
>
> Thank you for your valuable feedback. We apologize for any confusion caused by our previous statement. We have revised the statement to better reflect the intended meaning:
> 1. "In practice, individuals commonly use natural language to convey the  distinctive features of objects, thereby aiding in the differentiation of the discussed object from others."
> 2. We have revised the definition of IR to more closely align with the broader perspective in Manning, 2009.
> 3. We have corrected the citation of AdamW in the revised version.
>
> [1] Chang, Wei-Cheng, et al. "Pre-training tasks for embedding-based large-scale retrieval."
>
> [2] Lin, Sheng-Chieh, Jheng-Hong Yang, and Jimmy Lin. "In-batch negatives for knowledge distillation with tightly-coupled teachers for dense retrieval."
>
> [3] Gou, Jianping, et al. "Knowledge distillation: A survey."
>
> [4] Ren, Ruiyang, et al. "A thorough examination on zero-shot dense retrieval."

---

> ### Author Response · Authors · 2023-11-15
> **Response to Reviewer LJaD (Part 2/2)**
>
> **Q1: What the authors consider to be the difference between disentanglement and grounding in this case?**
>
> Thank you for the insightful question.
> - Disentangled representations, originating from Bengio's early work`[5,6]`, is defined as a property of "good" representations for machine learning, which focuses on the representations and the representation learning approach.
> - Grouding`[7]` generally refer to linking abstract knowledge (often in textual form) to tangible real-world examples, enabling AI systems to comprehend and interact with their environments. Grounding emphasizes practical applications and task-oriented solutions.
> - Disentangled representations benefit grounding tasks and a variety of other applications by improving interpretability, controllability, debuggability. For instance, both CLIP and VDR systems can identify relevance between images and text. While VDR employs disentangled representations, it offers a more nuanced perspective by pinpointing specific elements (tokens) that facilitate the matching process.
>
> **Q2: The authors claim that VDR outperforms the baselines with a significant margin, can the authors give the results of a significance test that backs up this claim?**
>
> We understand the importance of substantiating claims with empirical evidence and appreciate the opportunity to further clarify our findings.
>
> In our submission, we have been careful to specify conditions such as "with comparable model size and training costs" when referring to the primary baselines. When comparing VDR with advanced baselines, we have used terms like "comparable" or "fair" performance to avoid overstatements.
>
> Our evaluation adheres to well-established retrieval evaluation standards common in both academic and industrial research. Specifically, VDR exhibits an average improvement of +8.7 over DPR and +2.2 on SPLADE on NDCG@10 across 12 datasets. This margin of improvement is noteworthy, especially considering the VDR and DPR have similar parameter counts and identical training pipelines.
>
> To add depth to our analysis, we have responded to relevant questions from other reviewers that further elucidate our method's robustness. For instance, in our response to Reviewer fuH7 Q1 ([Link](https://openreview.net/forum?id=ZlQRiFmq7Y&noteId=FrI5Q8rbRr)), we have shown the stability of VDR with increasing vocabulary size. Additionally, in our reply to Reviewer LxNZ Q2 ([Link](https://openreview.net/forum?id=ZlQRiFmq7Y&noteId=O0mUHYTe37)), we provided insights into the underlying reasons for VDR's superior performance compared to DPR.
>
> We hope these togather can give your comprehensive view and address your concern. We are open to further discussion and engage into more dicussion to address your concerns.
>
> [5] Bengio, Yoshua. "Learning deep architectures for AI."
>
> [6] Bengio, Yoshua, Aaron Courville, and Pascal Vincent. "Representation learning: A review and new perspectives."
>
> [7] Chandu, Khyathi Raghavi, Yonatan Bisk, and Alan W. Black. "Grounding'Grounding'in NLP."

---

> ### Comment · Reviewer_LJaD · 2023-11-17
> **Thanks for the reply and some follow up questions**
>
> Thanks for your reply and additional explanations. A couple of follow up questions:
>
> Re W1(1): I am happy with this explanation, and I think it would be good to add a paragraph to the paper that explains in more detail how 'advanced' is defined. Maybe you can add parameter counts / computational cost to the results table.
>
> Re W1(2): I indeed read the additional explanation in Appendix H, but this still leaves me with a number of questions that are unaddressed, for example: (1) What platform was used? (2) How were the human raters selected? (3) participants are asked which set of tokens described the image better -- what was done to avoid position bias in this question? (4) In the second part of the human eval, did participants asses the same 20 images, or did everyone rate a different image? (5) What was done to compute interrupter agreement? Etc.
>
> Re Q2: My question was specifically targeted towards the claim that the scores were significant. Significance testing actually *is* a well-established practice within the IR community. Just some examples from SIGIR 2023 that report NDCG@k and report the results of significance testing:
> - https://arxiv.org/pdf/2305.01522.pdf
> - https://dl.acm.org/doi/pdf/10.1145/3539618.3591683
> - https://dl.acm.org/doi/pdf/10.1145/3539618.3591702
> - https://arxiv.org/pdf/2305.00319.pdf
> etc.

---

> ### Author Response · Authors · 2023-11-20
> **Response to follow up questions**
>
> We appreciate the reviewer's feedback, as it has been instrumental in enhancing the quality of our research. We are pleased to offer further clarifications below.
>
> > It would be good to add a paragraph to the paper that explains in more detail how 'advanced' is defined.
>
> Thank you for the suggestion! We have elaborated on the definition of advanced baselines in Appendix J and referenced it in Section 4.4 for clarity. Additionally, we've included details about parameter amount and training specifics. While computational costs are challenging to standardize due to varying infrastructures and preprocessing pipeline, we've provided our training time and device information to support future research comparison and replication efforts.
>
>
> **W1(2): Issues related to Evaluations.**
>
> > This still leaves me with a number of questions that are unaddressed, for example...
>
> (1) Our evaluation did not rely on a specific, pre-existing platform. Instead, we shared the evaluation samples and forms online directly with the raters.
>
> (2) Our raters were university students recruited with diverse backgrounds, which was to minimize bias and to reflect a broad spectrum of perspectives in the evaluation.
>
> (3) To mitigate potential position bias, we have randomized the order of the manually-selected top-5 tokens from VDR and also the presentations order of the two methods.
>
> (4) The participants evaluated the same 20 images in both phases (satisfaction and comparison). This is designed to ensure their familiarity for effective comparison, enhancing the reliability of their comparative assessments.
>
> (5) Regarding inter-rater (interrupter) agreement. Due to our budget limitations and the goal to cover a wide array of images, each image was assessed by a single rater. While this approach limits the application of standard inter-rater reliability metrics, we believe that the two-phase evaluation process, combined with the diversity of rater backgrounds, offers a multi-faceted perspective on the evaluations. We acknowledge this as a limitation in our study and suggest it as an area for future research with increased budget.
>
> These details are further elaborated in Appendix H of our submission. We hope this additional information addresses your concerns.
>
> **Q2: Can the authors give the results of a significance test that backs up this claim?**
>
> Thank you for your valuable feedback and for kindly sharing the examples to us. In response to your concerns, we have conducted comprehensive significance testing to reinforce the robustness of our work. We have included detailed analyses in Appendix K of our paper.
>
> Briefly, we conducted two-sided t-tests comparing VDR against baselines with identical parameter counts and training pipelines. Below tables are the results of significance tests where $\triangle$ means the improvement is significant with p<0.01. Our results demonstrated significant improvements (p<0.01) in the majority of text-to-text retrieval datasets and across all recall measures for cross-modal retrieval.
>
> |               | DPR   | VDR$_{t2t}$      |
> |---------------|-------|------------------|
> | ArguAna       | 40.8  | 48.8$\triangle$  |
> | Climate-FEVER | 16.2  | 18.1$\triangle$  |
> | DBPedia       | 30.4  | 37.6$\triangle$  |
> | FEVER         | 63.8  | 74.8$\triangle$  |
> | FiQA          | 23.7  | 29.3$\triangle$  |
> | HotpotQA      | 45.2  | 68.4             |
> | NFCorpus      | 26.1  | 32.7$\triangle$  |
> | NQ            | 43.2  | 45.8$\triangle$  |
> | SCIDOCs       | 10.9  | 15.4$\triangle$  |
> | SciFact       | 47.4  | 67.6$\triangle$  |
> | TREC-COVID    | 60.1  | 69.0             |
> | Touché-2020   | 22.1  | 27.7             |
>
>
> | Model       | MSCOCO i2t R@1 | R@5 | R@10 | t2i R@1 | R@5 | R@10 | F30k i2t R@1 | R@5 | R@10 | t2i R@1 | R@5 | R@10 |
> |-------------|-|-|-|-|-|-|-|-|-|-|-|-|
> | CLIP-BERT   | 23.9 | 47.8 | 60.3 | 13.6 | 33.8 | 45.1 | 44.1 | 71.2 | 80.7 | 27.8 | 54.7 | 65.9 |
> | VDR$_\mathrm{cm}$ | 30.9$\triangle$ | 54.5$\triangle$ | 65.4$\triangle$ | 17.4$\triangle$ | 38.1$\triangle$ | 49.7$\triangle$ | 51.0$\triangle$ | 79.3$\triangle$ | 86.7$\triangle$ | 32.4$\triangle$ | 60.1$\triangle$ | 70.7$\triangle$ |

---

### Official Review · Reviewer_fuH7 · 2023-11-07

**Soundness:** 3 good
**Presentation:** 3 good
**Contribution:** 3 good
**Rating:** 8
**Confidence:** 3

**Summary:**

1. The paper presents a novel approach for learning disentangled representations for text-to-text and text-to-image retrieval by leveraging an LM vocabulary as the distinct units of latent representation.

2. Concretely, the paper proposes casting both the query representation and the passage representations to a |V| dimensional space, with each value in the vector indicating the token's relevance to the input. This is then passed through a gating function to obtain the final (sparse) representation for each input. The models are trained end to end in a contrastive learning framework.

3. In order to extend the approach for multimodal retrieval scenarions, the authors introduce the following modifications:

3.1 Leveraging elu1p + top-k instead of ReLU to control for the number of sparse tokens in the representation and ensure the number of activated units are deterministic.

3.2 A non-parametric entry loss between the BoW representation of the query and the encoded representation of the data, thereby encouraging the model to avoid learning irrelevant co-activations.

3.3 In order to encourage activation of tokens that are in-frequent, the authors propose a contrastive mask, that activates a fixed fraction of activations within a batch, but discards their contribution while computing similarity between the positive samples; thereby encouraging the units to participate in the loss.

4. The proposed method demonstrates strong performance on both text-to-text retrieval (when trained on MS MARCO and tested on BEIR), as well as multimodal retrieval (when trained on YFCC15M and tested on ImageNet, COCO Captions, and Flickr30k).

4.1 In addition to that, for text only retrieval, leveraging a BOW representation of the query (thereby removing the need for a query encoder) achieves good retrieval performance while being up to 10x faster.

4.2 For the scenario of image disentanglement, based on human evaluation, for 92% cases the annotators found the disentangled representations to be a satisfactory representation of the image, compared to 85% when using a state-of-the-art image captioning model

**Strengths:**

1. The paper is well presented, I quite enjoyed reading it. The authors do a good job of presenting the approach in a clear manner. The idea of using the vocabulary from the tokenizer of a pre-trained model as the discrete latent units is quite neat.
2. The proposed modifications (w.r.t the elu1p + top-k activation, the non parametric entry loss and the contrastive mask) are well motivated.
3. The paper presents strong results for both text-to-text and text-to-image retrieval. For text-to-text, the results using non-parametric inference are particularly surprising and impressive, and potentially is generally applicable for tasks requiring fast inference. For the text-to-image retrieval scenario, both human evaluation as well as the qualitative analysis presented does establish the core premise of the paper of being able to leverage a text vocabulary for learning discrete latent representations.

**Weaknesses:**

1. Since the latent representations presented are tied to an underlying vocabulary, that inherently limits the kind of representations that the model can potentially learn. Because most of the modern day tokenizers are learned based on statistical properties, especially for models with larger vocabulary sizes (where each unit itself may not represent a meaningful entity), the representations learned may not be interpretable. The authors do showcase this limitation in one of the failure cases in the Appendix (w.r.t the tokenization of the word Giraffe).

2. The human evaluation setup for BLIP captioning is artificially detrimental to the image-captioning model. Specifically, the evaluators extracted 5 tokens from the caption to best reflect the image. However, the criterion of using top-k discrete tokens is a limitation of the proposed method and thus should not be imposed on the image captioning model. At the very least, using the complete caption should also be used (potentially as an upper bound for the task of which set of tokens better describe the image).

**Questions:**

1. How stable is the proposed approach to increasing the vocabulary size ? Concretely, would the performance be impacted if the underlying text encoder was XLM-Roberta (with a vocabulary of 250k)?
2. On a similar line of thought, what do the authors think about the feasibility of being able to decouple the vocabulary of the latent vector and the vocabulary used by the query encoder ? Concretely, would it be possible to have an XLM-Roberta encoder with the latent vectors being represented by a BERT vocabulary ?
3. For the text-to-text retrieval scenario, does the non-parametric inference still work without using the non-parametric entry loss ?
4. Would it be possible to elaborate more on how the analysis for Figure 4 (b) was done ? Is it considering only the patch embeddings for the max pooling operation in the DST head ?

Minor Typographic issues
Abstract: Our approach employ a bi-encoder -> our method employs a bi-encoder
Abstract: Moreover, The result -> Moreover, the result
Page 2: real-world data are -> real world data is
Page 4: to the tokens exist -> to the tokens that exist

---

> ### Author Response · Authors · 2023-11-15
> **Response to Reviewer fuH7**
>
> Thank you for your insightful feedback and an excellent summarization of our work! We address your comments and questions below.
>
>
> **W1: Unit itself may not represent a meaningful entity.**
>
> While our analysis show  most tokens in BERT vocabulary are interpretable, we acknowledge that this may not be universally applicable to all tokenizers. In cases where most tokens are not individually meaningful, we suggest pre-defining a set of textual concepts and measuring the data-concept relevance at a coarse-level, as conventional retrieval does.
>
>
> **W2: Human evaluation with complete BLIP caption.**
>
> Thank you for your valuable suggestion. We conducted an assessment on complete BLIP captions on the same sample set, and have detailed it in Appendix H. The evaluation revealed a 76% satisfaction rate, compared to 85% for the top-5 tokens manually selected from these captions. This discrepancy likely arises because BLIP captions, typically 2-10 words long, are succinct, making the top-5 tokens effective for summarizing core ideas. Furthermore, complete captions, while thorough, can risk in introducing specific but inaccurate details (like location and quantity), leading to a reduction in overall satisfaction.
>
>
> **Q1: How stable is the proposed approach to increasing the vocabulary size?**
>
> We experiments on mBERT (bert-base-multilingual-uncased) encoder with vocabulary size of 110k, as detailed in Appendix E. We observed a minor performance drop compared to BERT encoder (42.6 vs. 44.5). This slight decline is likely due to the gap between multilingual encoder and English-focused tasks but not the increased vocabulary size itself, as VDR consistently outperforms DPR by about 23% in both type of encoders. This finding align with our analysis in Section 3.3. The "silver bullet" here is that pretrained masked language models can provide a well-established gating distribution for subsequent dimension-wise supervision, which remains effective irrespective of vocabulary size changes.
>
> | |DPR(mBERT)|VDR(mBERT)|DPR(BERT)|VDR(BERT)|
> |-|--|---|---|---|
> |ArguAna|37.2|46.7|40.8|48.6|
> |Climate-FEVER|14.6|14.7|16.2|17.6|
> |DBPedia|31.3|36.4|30.4|39.0|
> |FEVER|62.7|69.7|63.8|74.0|
> |FiQA|21.6|27.5|23.7|28.8|
> |HotpotQA|45.6|63.1|45.2|65.5|
> |NFCorpus|25.2|33.3|26.1|33.0|
> |NQ|43.2|43.8|43.2|47.2|
> |SCIDOCs|10.7|14.3|10.9|15.3|
> |SciFact|48.2|66.7|47.4|67.3|
> |TREC-COVID|57.3|66.7|60.1|67.8|
> |Touché-2020|21.9|27.7|22.1|29.8|
> |Avg|34.9|42.6|35.8|44.5|
>
>
>
> **Q2: Would it be possible to have an XLM-Roberta encoder with the latent vectors being represented by a BERT vocabulary?**
>
> The reviewer raises an interesting point! Integrating an XLM-Roberta encoder with BERT vocabulary involves training a new linear projection in the DST head. This setup parallels the challenges encountered in cross-modal scenarios (Section 3.3). The feasibility varies, depending on whether we have a viable gating to start with:
> 1. Both Q and P side have well-established gating distribution (e.g., text-to-text senario). Effective supervision for downstream task optimization are achievable.
> 2. Only text-side has well-established gating distribution (e.g., cross-modal senario). Substantial training (more data, larger batches) and enhancement (contrastive mask) is necessary for valid supervision.
> 3. Neither side has well-established gating distribution (e.g., Appendix C). The bi-encoder struggle to conduct valid supervision and likely fail to learn useful representations.
>
> Therefore, substituting only one BERT-based encoder with XLM-Roberta is feasible. However, this may forfeit the advantages of a pre-trained MLM model, and a substantial training cost would be required to train this projection from scratch.
>
>
> **Q3: Does the non-parametric inference still work without using the non-parametric entry loss?**
>
> Our ablation study (Appendix I) reveals that while the non-parametric inference remains functional without this loss (denoted as "w/o bow" below), there is a noticeable decline in performance in NDCG@10 from 44.6 to 40.3.
>
> ||VDR|VDR(w/o bow)|VDR_a|VDR_a(w/o bow)|
> |---|:-------:|:------:|:------:|:-------:|
> |ArguAna|48.6|43.5|48.8|41.4|
> |Climate-FEVER|17.6|18.4|18.1|15.5|
> |DBPedia|39.0|38.3|37.6|34.4|
> |FEVER|74.0|73.7|74.8|70.7|
> |FiQA|28.8|28.1|29.3|25.3|
> |HotpotQA|65.5|64.5|68.4|63.2|
> |NFCorpus|33.0|33.2|32.7|32.3|
> |NQ|47.2|44.9|45.8|38.8|
> |SCIDOCs|15.3|14.7|15.4|14.0|
> |SciFact|67.3|65.7|67.6|66.0|
> |TREC-COVID|67.8|68.0|69.0|60.7|
> |Touché-2020|29.8|29.4|27.7|21.4|
> |Avg|44.5|43.5|44.6|40.3|
>
>
> **Q4: Would it be possible to elaborate more on how the analysis for Figure 4 (b) was done?**
>
> Your understanding is correct! We only aggregate the selected patch representations for the max pooling rather than pooling across all patches in DST head.
>
>
> **Q5: Typos issues.**
>
> Thank you for catching these! We have revised all the typos pointed out by the reviewers, and will continue to proof-read to further improve the manuscript.

---

> ### Author Response · Authors · 2023-11-20
> **Gentle Reminder**
>
> Dear Reviewer fuH7,
>
> We sincerely appreciate the time and effort you have dedicated to reviewing our paper. Your insightful questions and valuable feedback have greatly contributed to the enhancement of our research.
>
> In response to your valuable feedback, we have diligently worked on a rebuttal that addresses concerns you raised. As the discussion period is approaching its conclusion in two days, we would greatly appreciate your review of our rebuttal at your earliest convenience. Should there be any further points that require clarification or improvement, please know that we are fully committed to addressing them promptly.
>
> Thank you once again for your invaluable contribution to our work.
>
> Warm Regards,
>
> The Authors

---

### Author Response · Authors · 2023-11-15
**General Response to All Reviewers**

We thank all the reviewers for their thoughtful and constructive feedback! We are pleased they found the work is well-presented, the problem we address is important, and the modifications we proposed are well motivated.

Here, we briefly summarize the updates made to the manuscript in response to reviewer feedback.

***Text Changes***: The following updates have been made to improve clarity and readability.
- **Section 1**: We revised certain sentences to avoid confusion, and addressed some typographical errors.
- **Section 2**: We expanded the definition of information retrieval to align with (Manning, 2009), in response to reviewer LjaD's suggestion.

***Additional Experiments***: The following experiments have been conducted to address reviewers' concerns.
- **Appendix E**: We've added a new subsection that analyzes the stability of our method when the vocabulary size is increased.
- **Appendix H**: The human evaluation section has been augmented with a baseline using the full BLIP caption along with more details and analyasis.
- **Appendix I**: An ablation study has been introduced in this section to assess the different components of VDR in text-to-text retrieval tasks.
- **Appendix J**: We delve into the definition of advanced baselines and discuss the fairness in the baseline distinction.
- **Appendix K**: We present significance tests to demonstrate the improvements of our method are statistically significant.

To enhance the reading experience, we have summarized the main concerns addressed for each reviewer below.

***Response to Reviewer fuH7*** ([Link](https://openreview.net/forum?id=ZlQRiFmq7Y&noteId=FrI5Q8rbRr))
- Q1: Stableness with increasing the vocabulary size.
- Q2: Feasibility of using other encoder along with BERT vocabulary.
- Q3: Effect on non-parametric inference without non-parametric entry loss.
- Q4: Elaboration on the setup of Figure 4(b).

***Response to Reviewer LJaD*** ([Link](https://openreview.net/forum?id=ZlQRiFmq7Y&noteId=HgBjwPv8Ff))
- W1(1): Definition of 'advanced' baseline and its fairness.
- W1(2): Issues related to Evaluations.
- Q1: Discussion on difference between disentanglement and grounding.
- Q2: Significance test of improvements.


***Response to Reviewer LxNZ*** ([Link](https://openreview.net/forum?id=ZlQRiFmq7Y&noteId=O0mUHYTe37))
- W1: Demonstration to enhance explainability.
- Q1: Ablation study on elu1p and Relu activation.
- Q2: Intuition on why VDR outperform DPR.

---

### Meta-Review · Program_Chairs · 2023-12-13

**Metareview:**

The paper presents a text-to-text and text-to-image retrieval approach that leverages LM vocabulary representations as the latent space.  The approach is innovative.  The work is sound.  And the reviewers are in favour of acceptance.

**Justification For Why Not Higher Score:**

The reviewers pointed out some problems in the evaluation, especially the human evaluations,  and in the baselines used which are stripped from "advanced" enhancements. They also point out that that the disentanglement is not shown beyond retrieval tasks.

**Justification For Why Not Lower Score:**

As written in the metareview.

---

### Decision · Program_Chairs · 2024-01-16

Accept (spotlight)